# Fluctuating reproductive isolation and stable ancestry structure in a fine-scaled mosaic of hybridizing *Mimulus* monkeyflowers

**Matthew C. Farnitano**[1]*, **Keith Karoly**[2], **Andrea L. Sweigart**[1]

1 Department of Genetics, University of Georgia, Athens, Georgia, United States of America,
2 Department of Biology, Reed College, Portland, Oregon, United States of America

* mattfarnitano@gmail.com

## Abstract

Hybridization among taxa impacts a variety of evolutionary processes from adaptation to extinction. We seek to understand both patterns of hybridization across taxa and the evolutionary and ecological forces driving those patterns. To this end, we use whole-genome low-coverage sequencing of 458 wild-grown and 1565 offspring individuals to characterize the structure, stability, and mating dynamics of admixed populations of *Mimulus guttatus* and *Mimulus nasutus* across a decade of sampling. In three streams, admixed genomes are common and a *M. nasutus* organellar haplotype is fixed in *M. guttatus,* but new hybridization events are rare. Admixture is strongly unidirectional, but each stream has a unique distribution of ancestry proportions. In one stream, three distinct cohorts of admixed ancestry are spatially structured at ~20-50m resolution and stable across years. Mating system provides almost complete isolation of *M. nasutus* from both *M. guttatus* and admixed cohorts*,* and is a partial barrier between admixed and *M. guttatus* cohorts. Isolation due to phenology is near-complete between *M. guttatus* and *M. nasutus.* Phenological isolation is a strong barrier in some years between admixed and *M. guttatus* cohorts, but a much weaker barrier in other years, providing a potential bridge for gene flow. These fluctuations are associated with differences in water availability across years, supporting a role for climate in mediating the strength of reproductive isolation. Together, mating system and phenology accurately predict fluctuations in assortative mating across years, which we estimate directly using paired maternal and offspring genotypes. Climate-driven fluctuations in reproductive isolation may promote the longer-term stability of a complex mosaic of hybrid ancestry, preventing either complete isolation or complete collapse of species barriers.

## Author summary

Hybridization between species can create genetic novelty and promote adaptation, but can also erode species barriers and dilute genetic diversity. Climatic variation likely impacts the extent and eventual outcomes of hybridization, but these impacts are difficult to predict. We use population-scale genomic sequencing of hybridizing *Mimulus*

**Data availability statement:** All Illumina data generated for this project are archived at the Sequence Read Archive (https://www.ncbi.nlm.nih.gov/sra) under BioProject PRJNA1226178. Phenological data, sample information, and other metadata are archived at the DRYAD digital repository (DOI: https://doi.org/10.5061/dryad.kd51c5bhm). All analysis scripts are available on GitHub (github.com/mfarnitano/CAC_popgen).

**Funding:** ALS was supported by National Science Foundation award DEB1856180. MCF was supported by the National Institutes of Health Predoctoral Training Grant 5T32GM007103, the Society for the Study of Evolution GREG R.C. Lewontin Award, and a Doctoral Dissertation Improvement Grant (DDIG) from The Plant Center at the University of Georgia. The funders had no role in study design, data collection and analysis, decision to publish, or preparation of the manuscript.

**Competing interests:** The authors have declared that no competing interests exist.

monkeyflowers to better understand the influence of climatic variation on hybridization. We find evidence of hybridization in multiple populations, but the distribution of hybrid ancestry varies. In one population, groups of distinct hybrid ancestries are clustered in close proximity to each other. Isolation between these ancestry groups is driven in part by variation in self-fertilization and flowering time. However, some years have more overlap in flowering time than others, potentially driven by the timing of water availability. We see that hybrid ancestry is remarkably stable across a decade of measurements, despite fluctuations in the degree of isolation. Paradoxically, fluctuations in isolation may allow hybrid ancestry clusters to persist by preventing complete erosion of species barriers while still allowing some gene exchange. Climate change is expected to increase the variability of climatic factors such as precipitation and heat events; our study demonstrates one way these fluctuations could impact species.

## Introduction

Hybridization can have a wide range of impacts on species, depending on both the success of hybrid individuals and their degree of reproductive isolation from progenitors. At one extreme, unfit hybrids can be an evolutionary dead-end [1–6]. Conversely, hybrids may freely and successfully mate with both progenitors, eroding differences between species until a single undifferentiated population remains [7–9]. Another possibility is that hybrids are successful but become strongly reproductively isolated from progenitors, forming a new, independent lineage [10–13]. Many cases fall between these extremes, with partial but incomplete reproductive barriers [14–17].

Partial reproductive isolation allows for ongoing gene flow between species (introgression) without collapse into a single lineage. Plant biologists have long understood the importance of hybridization among members of a 'syngameon', a group of species that exchange genes but maintain distinctiveness [18,19]. In the genomic era, introgression has been detected throughout the tree of life, suggesting that partial reproductive isolation is common [20,21]. Introgression plays an important role in numerous evolutionary processes such as adaptation [22,23], niche expansion [24,25], evolutionary rescue [26,27], extinction [28,29], and invasion [30,31]. But the dynamics of introgression are varied. Hybridization can be rare and transient, followed by backcrossing within a few generations to leave a signature of introgression at just a few genetic loci [32–34]. In other cases, hybridization produces a persistent swarm of intermating hybrids, which then have the potential to interact more extensively with progenitor species [35–38]. The coexistence of a partially isolated hybrid swarm alongside progenitor species provides an opportunity for significant adaptive introgression, since hybrids can act as a bridge for gene flow between progenitors [39,40]. However, hybridization can also introduce maladaptive alleles or allele combinations [1,41–43] that an admixed population must contend with [37,44–46]. Theory suggests that intermediate levels of reproductive isolation may, under certain circumstances, be an evolutionarily stable state rather than simply a transition state on the route to complete speciation [47]. But hybridization outcomes appear to be highly contingent on environmental and genetic conditions [35,48–50]. More empirical work is needed to understand what conditions lead to persistent partial reproductive isolation in hybrid populations.

One factor that could lead to partial reproductive isolation is spatial heterogeneity. When multiple distinct microhabitats are available, divergent directional selection can maintain multiple ecotypes in parallel niches [51–53]. Selection on locally favored alleles in these microhabitats can act as a premating reproductive barrier, reducing gene flow between ecotypes by

eliminating migrants [54–57]. On the other hand, dispersal between these microhabitats could erode both local adaptation and reproductive isolation between ecotypes, leading to collapse into a single gene pool [7,58,59]. Within this pool, multiple microhabitats may instead promote a diversity of alleles through balancing selection [60,61]. The relative importance of divergent selection and balancing selection will depend on the scale and frequency of dispersal relative to the scale of environmental heterogeneity [62–65]. In some cases, dispersal may be sufficient to prevent complete isolation but not strong enough to erode differences completely. In fact, gene flow is often detected in sympatry even when premating reproductive barriers are strong [66–68]. Furthermore, hybrids may establish their own niche in intermediate or underutilized microhabitats, in which persistence is subject to the same balance of migration and selection [11,50,69–71].

Environmental heterogeneity over time, both within and across years, might also contribute to the maintenance of partial reproductive isolation. Differences in the timing of mating throughout the year can isolate groups [72–74]. If reproductive phenology is dependent on environmental cues, then variation in those cues across years can modulate the strength of temporal isolation [75–77]. Also, variation in selective pressures across years can lead to fluctuating selection that maintains a diversity of genotypes [78–81]. Within a hybridizing population, this can result in selection against hybridization in some years but not in others [82,83].

Understanding and predicting the effects of environmental variation on hybridization and reproductive isolation requires tracking hybrid populations across time. Multi-year studies have given us important insights into hybridization, demonstrating directional shifts in composition over time [84], shifts across space [85], stability over time [86,87], and fluctuations in the strength of species barriers [9,82,88]. Still, we know little about the environmental and genetic circumstances driving these different outcomes.

With this in mind, we turn to a previously identified hybridizing population of *Mimulus guttatus* and *Mimulus nasutus* monkeyflowers, at Catherine Creek (CAC) just north of the Columbia River Gorge in Washington, USA (Fig 1A). *M. nasutus* diverged from an *M. guttatus* progenitor ~200KYA, expanding to share much of the *M. guttatus* range across the western United States, where the two hybridize in multiple locations of secondary contact [89]. At Catherine Creek and the surrounding area, both species occupy a series of ephemeral seeps, where they co-occur at small spatial scales [90]. Previously, *M. guttatus* collections in 2012 from two parallel streams at Catherine Creek (CAC_Stream1 and CAC_Stream2, Fig 1A) showed levels of *M. nasutus* genomic ancestry ranging from near 0 to approximately 50%, indicating a history of hybridization followed by backcrossing to the *M. guttatus* parental population [89,90].

Despite this evidence of hybridization, several prezygotic reproductive barriers have been documented between *M. guttatus* and *M. nasutus*. One major source of isolation is mating system: *M. guttatus* is primarily outcrossing (though self-compatible), with large showy flowers visited by a variety of bee pollinators [91–93], while *M. nasutus* is predominantly selfing, with small and often cleistogamous flowers that self-pollinate prior to opening [94,95]. In addition, the species have phenological differences closely tied to water availability and drought escape: *M. nasutus* tends to be found on mossy rock outcrops that dry out more quickly, while the seepy microhabitats of *M. guttatus* stay wet later into the spring [57,90,96]. *M. nasutus* is better able to accelerate its life cycle to escape terminal drought, while the same drought stress causes a fitness loss in *M. guttatus* [57]. *M. nasutus* also flowers earlier in the season than *M. guttatus*, in part due to a shorter photoperiod requirement for flowering; this interspecific difference has been mapped to multiple large-effect genetic loci [97] and contributes to reproductive isolation in the wild by reducing phenological overlap [90]. But

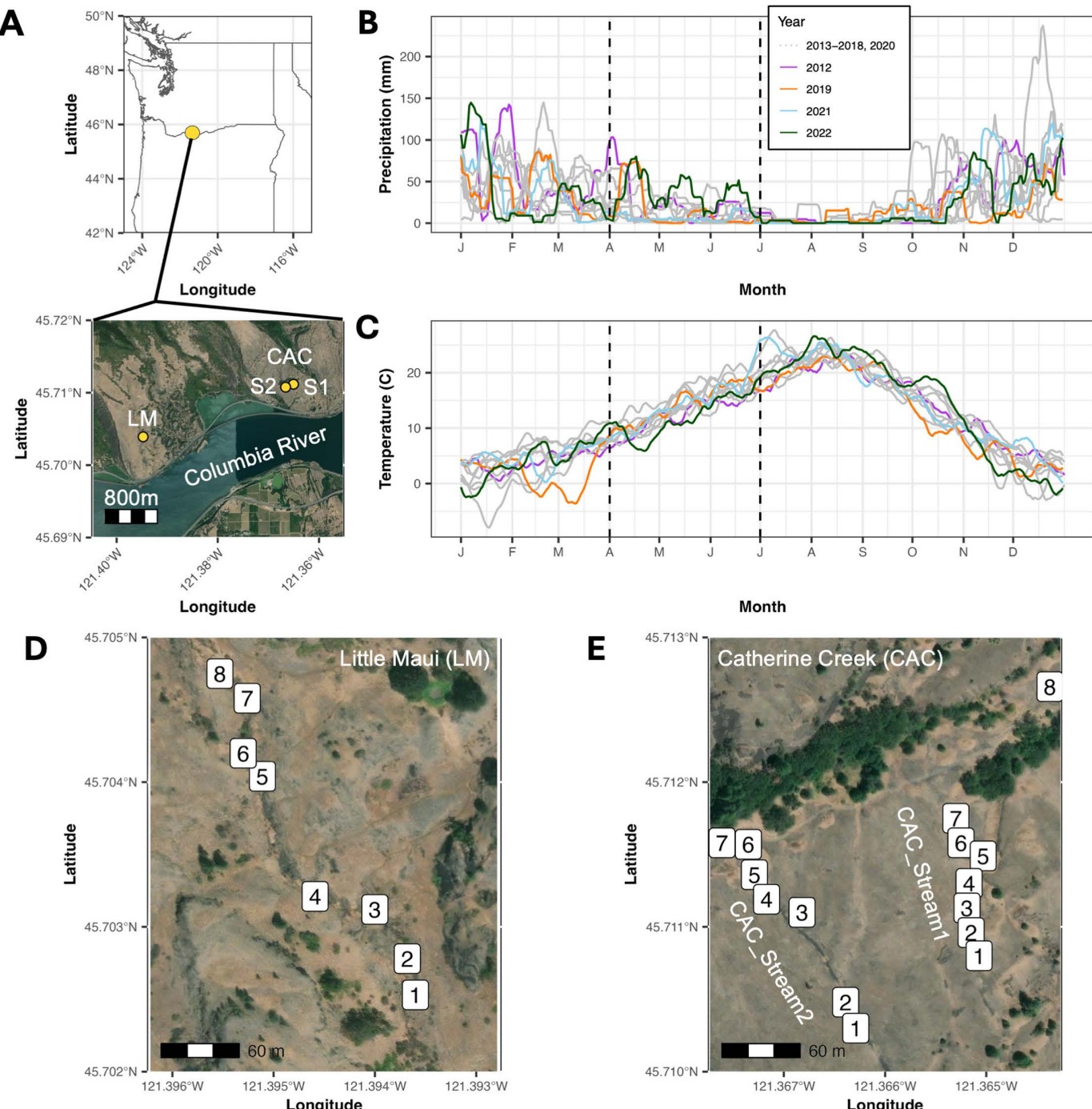

**Fig 1. Sampling locations and local environmental variation.** A) Location of three sampling locations in the Columbia River Gorge area, Washington, USA. LM=Little Maui stream, CAC=Catherine Creek site with two parallel streams, S1=CAC_Stream1 and S2=CAC_Stream2. Approximate distance between CAC_Stream1 and CAC_Stream2 is ~120m, between CAC and LM is ~2.5 km. B-C) Interpolated precipitation totals (B) and average temperatures (C) in 14-day sliding windows for a 4km grid square covering the CAC and LM sites, downloaded from PRISM [147]. Colored lines indicate years where we sampled for this study; other intervening years are colored in grey. Dashed vertical lines indicate extent of typical flowering season for CAC and LM *Mimulus,* from April to June. D-E) Sample plots within each of three sampled streams: (D) LM, (E, left) CAC_Stream2, and (E, right) CAC_Stream1. A 0.5mx0.5m square was placed at each plot for flower counts and collections. Exact square placement varied slightly across years, and not all plot locations were sampled in each year. State and country outlines were obtained from gadm.org using the R/geodata package [178]. Aerial maps were obtained from the National Agricultural Imagery Program, courtesy of the U. S. Department of Agriculture, Farm Service Agency [179].

phenology may also be responsive to fluctuating environmental conditions, and the amount of phenological overlap could depend on a number of other factors besides a simple photoperiod cue, such as the duration of flowering and the quantity of floral displays. The Catherine Creek area has highly variable precipitation (both in quantity and seasonal timing, Fig 1B), so we predict that heterogeneous water availability plays an important role in the persistence and isolation of these species and their hybrids. Other environmental variables, such as temperature (Fig 1C) might also impact selection and isolation within these species and their hybrids.

Here, we use a low-coverage whole-genome approach to sequence wild-growing individuals from an additional nearby site 2.5 km away from Catherine Creek, named Little Maui (LM), in order to compare the structure of hybrid ancestry across multiple independent hybrid contact zones. In addition, we sequence three additional years of wild-growing individuals from one focal stream at Catherine Creek (labeled CAC_Stream1), and two years of samples from a second parallel stream ~150m away (labeled CAC_Stream2) along with offspring germinated from seeds of these wild individuals. This multi-year dataset, spanning a decade when combined with previous data, allows us to ask whether the hybrid population at Catherine Creek is stable across years. Next, we address multiple possible contributors to reproductive isolation among ancestry cohorts within Catherine Creek: spatial segregation, mating system variation, and phenological isolation across the flowering season. Using offspring genotypes, we document how these factors influence the actual mating dynamics of the admixed population, and how these dynamics change across years. We develop hypotheses about how fluctuating environmental conditions over time might interact with spatial heterogeneity to create contrasting ancestry patterns in different contact zones.

# Results

## Introgression from *M. nasutus* in replicated contact zones

To explore patterns of genomic variation within and between sympatric populations, we used a low-coverage whole-genome approach to sequence a total of 458 wild *Mimulus* samples collected from three streams in the Columbia River Gorge area (Table 1 and Fig 1A, 1D, and 1E): 265 from CAC_Stream1 collected in 2019, 2021, and 2022; 133 from CAC_Stream2 collected in 2021 and 2022; and 61 from Little Maui (LM, 2.5 km from CAC) collected only in 2021.

**Table 1. Maternal and offspring samples sequenced for this study from three streams.**

| Year | Wild maternal samples | | | | | | Offspring | |
|---|---|---|---|---|---|---|---|---|
| | Raw | Filt. | *M. nasutus* (HI>0.95) | *M. guttatus* (HI<0.15) | admixed (HI 0.15-0.8) | *M. sookensis* | Raw | Filt. |
| **CAC_Stream1** | | | | | | | | |
| 2012* | | 23 | 3 | 4 | 16 | | | |
| 2019 | 153 | 125 | 29 | 38 | 58 | | 580 | 549 |
| 2021 | 18 | 17 | 0 | 13 | 4 | | 192 | 166 |
| 2022 | 133 | 122 | 19 | 48 | 55 | | 291 | 289 |
| **CAC_Stream2** | | | | | | | | |
| 2012* | | 52 | 4 | 40 | 8 | | | |
| 2021 | 14 | 11 | 0 | 7 | 4 | | 189 | 162 |
| 2022 | 130 | 122 | 10 | 34 | 67 | 11 | 334 | 329 |
| **LM** | | | | | | | | |
| 2021 | 64 | 61 | 3 | 2 | 56 | | 848 | 753 |

*2012 samples are from previously published data (90).

Filt. = Filtered for>=25,000 sites with ancestry assignments

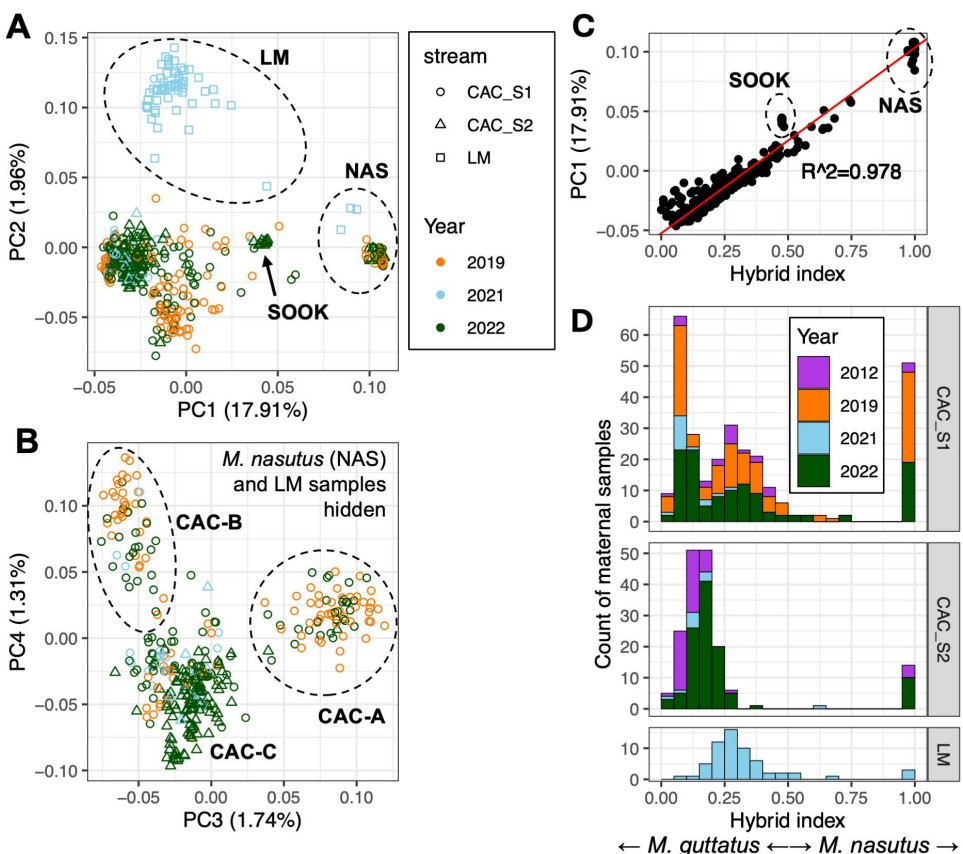

**Fig 2. Directional admixture shapes population structure in replicate streams.** A) Genomic PCA based on low-coverage genotype likelihoods at 19,633 variant sites for 264 individuals from CAC_Stream1 (CAC_S1, sampled in 2019, 2021, and 2022), 133 individuals from CAC_Stream2 (CAC_S2, sampled in 2021 and 2022), and 61 individuals from LM (sampled in 2021 only). PC1 separates *M. nasutus* samples (NAS) from *M. guttatus* and admixed samples, while PC2 separates Little Maui (LM) samples from Catherine Creek (CAC) samples. B) PC3 and PC4 from the same genomic PCA, with Little Maui (LM) and *M. nasutus* (NAS) samples removed from the visualization to more clearly show differentiation in the other samples. PC3 differentiates an admixed group found primarily in plots S1_1 and S1_2 (cluster CAC-A: note that these plots were not sampled in 2021), while PC4 separates a less-admixed group found primarily in plot S1_5 (cluster CAC-B). The remaining cluster (CAC-C) includes individuals from both CAC streams, which are not differentiated by these PC axes. Variation within Catherine Creek is not structured by year. C) PC axis 1 correlates strongly with hybrid index (the proportion of *M. nasutus* genomic ancestry, determined by local ancestry inference). NAS=*M. nasutus* individuals, SOOK=11 individuals of *M. sookensis*, an allopolyploid species between a *M. guttatus*-like progenitor and *M. nasutus*, which has an expected 50/50 ancestry composition between the two species. D) Hybrid index, the proportion of sites with *M. nasutus* ancestry across the genome for each individual, is distributed differently in each replicate stream, but this pattern is consistent across years. Hybrid index of 0.0 indicates *M. guttatus*, 1.0 indicates *M. nasutus*. Data from 2012 are from [90]; note that hybrid index for these samples was obtained using a different method.

To explore genomic variation within and between streams, we performed a genomic PCA on these samples (Fig 2A and 2B). PC axis 1 (17.91% of variation) separates species by ancestry, showing a clearly differentiated *M. nasutus* group and a cloud of *M. guttatus*-like individuals with varying levels of hybrid ancestry (Fig 2A). This primary axis of variation is highly correlated with hybrid index (proportion *M. nasutus* vs. *M. guttatus* ancestry) as determined by local ancestry inference ($r^2$=0.978, Fig 2C). A separate NGSadmix structure analysis with K=2 is also highly correlated with hybrid index ($r^2$=0.979, S1 Fig), further supporting hybrid ancestry proportion as the major axis of genomic differentiation in these populations.

Additional PC axes reveal population structure associated with geography, both between and within streams. PC2 (1.96% of variation) cleanly separates LM from CAC samples (true for both admixed individuals around PC1 ~ 0 and to a lesser extent for *M. nasutus* around PC1 ~ 0.1, Fig 2A). PC3 (1.74%) and PC4 (1.31%) further differentiate three clusters of admixed individuals within Catherine Creek (Fig 2B: LM and *M. nasutus* samples hidden for clarity). Two of these three clusters contain only samples from one stream (CAC_Stream1) while the third contains all remaining samples from both CAC streams. Within CAC_Stream1, each study plot is largely confined to one of these three clusters (S1 Table), suggesting that these axes of differentiation reflect spatial structure within CAC_Stream1 at a scale of <300m. We find similar clustering within CAC_Stream1 when running a separate PCA of only CAC individuals with hybrid index <0.8 (i.e., excluding LM and *M. nasutus* individuals, S2 Fig). Notably, despite differentiation within CAC_Stream1, samples from the parallel stream CAC_Stream2 (~150 m away) are not differentiated from CAC_Stream1 nor do they form multiple clusters, at least in these first four PC axes (main PCA, Fig 1A and 1B, and CAC-only PCA, S2 Fig). Each PCA cluster contains samples from multiple sampling years and therefore multiple sequencing batches (see Materials and Methods), except the LM cluster because LM was only sampled in one year, indicating that year and associated batch effects do not account for the clustering.

Consistent with previous population genomic analyses [89,90], the distribution of hybrid index (HI, the proportion of the genome with *M. nasutus* ancestry) suggests mostly unidirectional backcrossing of hybrids with *M. guttatus* (i.e., a majority of admixed individuals have HI<0.5, Fig 2D). In fact, virtually all CAC and LM *M. guttatus*-like individuals have at least some detectable *M. nasutus* ancestry: out of 374 majority-*guttatus* samples, only 12 have HI<0.05, while only two of those have HI<0.01. The *M. nasutus* samples, in contrast, have little to no introgression: out of 74 majority-*nasutus* samples, 61 have HI>0.95 and 57 of those have HI>0.99. The remaining 13 have HI between 0.5 and 0.8, indicating that backcrossing with *M. nasutus* does occasionally happen. No individuals were sampled with HI between 0.8 and 0.95, suggesting that backcrossing to *M. nasutus* does not typically continue for multiple generations as it does with *M. guttatus*. Despite pervasive admixture, we detect no first-generation (F1) hybrids (S3 Fig). We did discover a group of 11 individuals from CAC_Stream2 with HIs near 0.5, which were determined to be the allopolyploid species *M. sookensis* (Figs 2C, S3 and S4, and Materials and Methods). These provide a useful check of our ancestry calling method, since they have an expected HI of 0.5 due to apparent fixed heterozygosity caused by two homeologous copies of each locus [98]; once identified, they were removed from further analyses of admixture and reproductive isolation. Of the remaining 27 individuals with HIs between 0.4 and 0.6, all have ancestry heterozygosity values less than 0.62 (S3 Fig), indicating they are second- or later-generation hybrids as opposed to new F1s (which should have heterozygosity near 1.0). Taken together, our finding that directional introgression composed of later-generation and backcrossed hybrids is replicated in all three sampled streams indicates admixture in secondary contact is not only common but follows predictable patterns across the landscape.

Given the admixture we detect in the nuclear genome, we asked whether organellar genomes also show signatures of introgression. Remarkably, we found that all 325 CAC and LM samples included in our chloroplast network analysis – whether *M. guttatus,* admixed, or *M. nasutus* - have a single chloroplast haplotype (Fig 3). A similar pattern is seen in the mitochondria (S5 Fig). This same haplotype (or close derivatives) is carried by *all* sampled *M. nasutus* and *M. sookensis* (an allopolyploid with a maternal *M. nasutus* origin) lineages, including collections spanning most of the two species' ranges [98]. In contrast, outside of CAC and LM, the haplotype is almost never found in *M. guttatus,* which has much higher

organellar diversity across its geographic range (Figs 3 and S5): of 18 previously sampled non-CAC/LM *M. guttatus* accessions, only two carry this CAC/LM haplotype. One of these two accessions is the only other *M. guttatus* sample in our dataset from a known sympatric site: DPR, which is ~1000 km south of CAC/LM in the Sierras of California and shows a signature

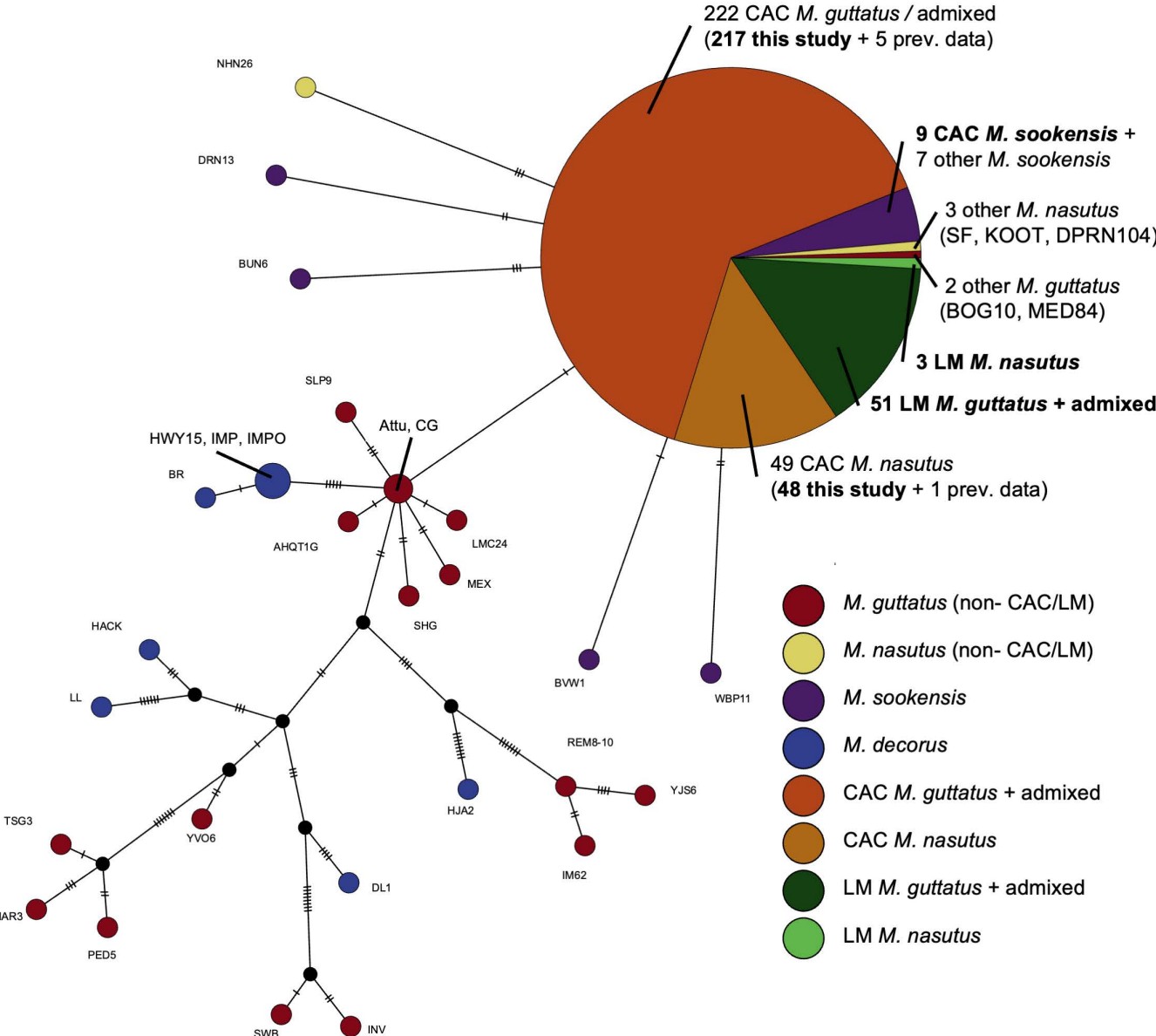

**Fig 3. Complete chloroplast capture of a *M. nasutus* haplotype in sympatric *M. guttatus*.** NJ-Net haplotype network of chloroplast variation built from 328 CAC and LM maternal samples sequenced in this study (bolded captions) and 47 previously sequenced samples (non-bolded captions) from the *M. guttatus* species complex, including *M. nasutus, M. guttatus, M. sookensis* (a polyploid with *M. nasutus* as maternal parent), and *M. decorus* (another close relative) [98], using 102 total variant sites (43 parsimony-informative). Each circle represents a unique haplotype, with the size of the circle proportional to the number of samples matching that haplotype. Haplotypes are connected according to sequence similarity, with tick marks along the connections representing the number of nucleotide changes separating each sequence. A single chloroplast haplotype is present in all maternal samples from Catherine Creek, including *M. guttatus,* admixed, *M. nasutus,* and *M. sookensis* samples. All other *M. nasutus* samples from across the range share this same haplotype or a close derivative, as do samples from *M. sookensis* (a polyploid with *M. nasutus* as maternal parent). *M. guttatus* haplotypes are more variable, with only two samples not from CAC or LM sharing the *M. nasutus* haplotype, at least one of which (DPR84) is from another sympatric site with known introgression.

of *M. nasutus* introgression [46,89]. We therefore infer that this haplotype is derived from *M. nasutus* and has been completely captured by sympatric *M. guttatus* through introgression at both CAC and LM sites (and, potentially, at other sympatric populations elsewhere in the range, such as DPR). This haplotype structure suggests that not a single *M. guttatus* individual in our sample is free from the effects of introgression.

### Patterns of ancestry at small spatial scales are persistent across years.

While we find substantial admixture in all three streams, ancestry proportions vary across the landscape (Fig 2D). In each stream, we see multiple distinct, though sometimes overlapping, peaks of ancestry, which we refer to as ancestry cohorts. All three streams have a clear *M. nasutus* cohort (HI>0.95). LM and CAC_Stream2 each have a second admixed cohort with majority-*M. guttatus* ancestry, but the two streams have different peak ancestries (HI=0.25-0.3 for LM and 0.1-0.2 for CAC_Stream2). Within CAC_Stream1, we see three cohorts: *M. nasutus,* a *M. guttatus*-like cohort centered around HI=0.05-0.1, and a more admixed cohort centered around HI=0.25-0.3. Where we have multiple years of sampling, the distribution of ancestry in each stream appears similar across years: in particular, all three cohorts (*M. guttatus,* admixed, and *M. nasutus*) in CAC_Stream1 are present in 2012, 2019, and 2022; note that *M. nasutus* at CAC were not sampled in 2021 because we arrived at the site later in the season that year. This year-to-year consistency both validates the reliability of our ancestry assignments and suggests that the three-cohort ancestry structure of CAC_Stream1 is stable across time, even though such structure is not found in other nearby streams.

We asked whether the multiple peaks of ancestry within CAC_Stream1 correspond to spatial structure at a smaller scale. The distribution of hybrid ancestry varies across plots, with clear spatial segregation between the three ancestry cohorts (Fig 4A). These differences are apparent on very fine scales: plots ~20m apart have distinct distributions of ancestry (i.e., plots S1_4, S1_5, and S1_6; Figs 1C and 4A). In contrast, LM and CAC_Stream2 appear to have less clear differentiation across space, though *M. nasutus* samples are clustered towards the lower plots in both (Fig 4B and 4C). The CAC_Stream1 spatial pattern of ancestry is remarkably consistent across years wherever we have multiple years of sampling (Fig 4A). This is true even comparing 2012 (which used a different sequencing methodology) to later years. There are a few exceptions: plot S1_7, for example, appears to have become more admixed between 2012 and 2019. But overall, differences in admixture proportion at small spatial scales are remarkably consistent across a decade of sampling in CAC_Stream1, contrasting with weaker spatial segregation in the other streams.

### Partial reproductive isolation between admixture cohorts due to mating system

We next sought to investigate potential reasons for the persistence of spatial ancestry structure at such fine scales. One possible cause is premating isolation driven by self-fertilization, so we used paired maternal and offspring sequence data within the two CAC streams to infer selfing rates for our samples. After filtering for sequence quality and maternal match, we inferred selfing status for 1045 offspring from 189 individual fruits across 151 maternal families (Table 2). We found that, as expected, *M. nasutus* samples are highly selfing: 0 of 43 offspring from *M. nasutus* maternal plants were inferred to be outcrossed. *M. guttatus* and admixed cohorts had a mix of selfing and outcrossing: selfing rates for *M. guttatus* individuals with HI<0.15 ranged from 0 to 0.217 depending on year and stream (Table 2) while more admixed individuals (HI=0.15-0.8) had similar selfing rates ranging from 0.026-0.365, Table 2). Selfing rate significantly increased with increasing *M. nasutus* ancestry, though this pattern is primarily

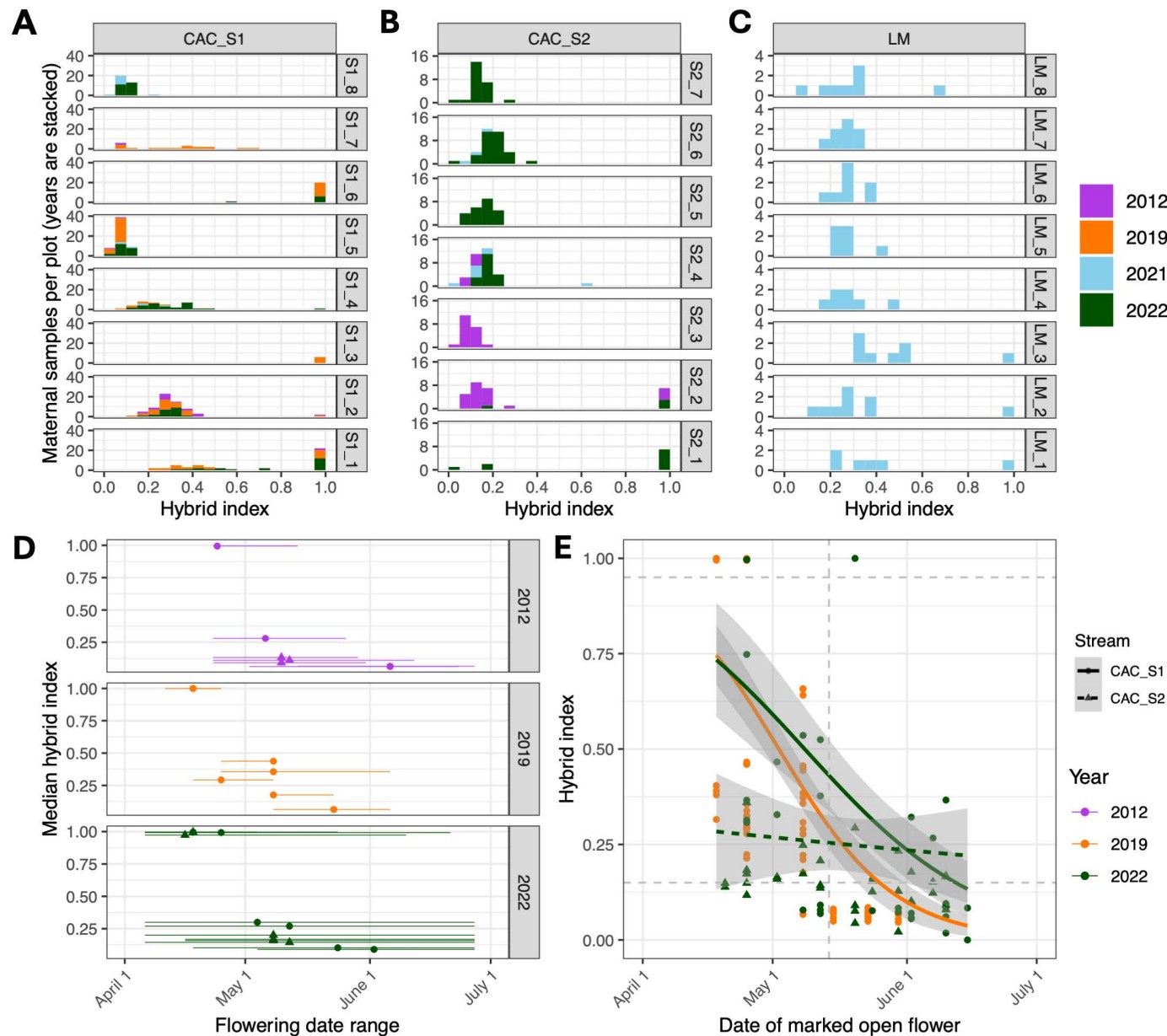

**Fig 4. Fine-scaled spatial and phenological structure across years.** A) Histogram of maternal samples sequenced from each of eight plots within CAC_Stream1 (CAC_S1), binned by hybrid index, with sampling years in stacked bars. Hybrid index of 0.0 indicates *M. guttatus*, 1.0 indicates *M. nasutus*. Plots vary in the distribution of *M. guttatus,* admixed, and *M. nasutus* individuals, but most have a consistent distribution across years. B) Histograms of hybrid index within each of seven plots from CAC_Stream2 (CAC_S2), sampled in 2012 (previous data), 2021 (limited sampling), and 2022. C) Histograms of hybrid index within each of eight Little Maui (LM) plots, sampled in 2021 only. D) Median flower date and range of first to last open flower dates for each census plot in CAC_Stream1 and CAC_Stream2, compared to median hybrid index of sequenced samples from that plot. Median flower date is associated with median hybrid index. Flowering season duration was longer for most plots in 2022 compared to previous years. E) Date of marked open flowers is associated with hybrid index of the marked individual for CAC_Stream2, but not for CAC_Stream1. Random sets of flowers were marked when open throughout the season and those individuals were later sampled for sequencing; these are not necessarily the first open flowers for an individual. Colored lines and grey intervals represent model fits and 95% confidence intervals from a beta GAM regression, using year and stream as additional factor variables.

driven by *M. nasutus* individuals: when *M. nasutus* (HI>0.8) families are excluded, selfing increases with ancestry in 2021 and 2022 but decreases with ancestry in 2019 (S2 Table and S6 Fig). Selfing rate also significantly varied across years, with 2022 having the highest selfing rates across all ancestry cohorts (Tables 2 and S2 and S6 Fig). There was no significant effect of stream (CAC_Stream1 vs. CAC_Stream2) on selfing rates (Tables 2 and S2 and S6 Fig). Within the *M. guttatus* and admixed cohorts, most selfed offspring were found alongside outcrossed offspring in the same fruit: 13% of fruits with at least two called offspring had a mix of selfed and outcrossed offspring (S3 Table). Of these mixed fruits, most were majority outcrossing with a mean selfing proportion of 37% (S3 Table). Multiple paternity within a fruit was also common: out of 155 fruits with at least two outcrossed offspring, 119 (77%) had more than one inferred pollen donor, with an average probability of 52.9% that two outcrossed offspring in the same fruit were half-siblings rather than full-siblings (S3 Table).

We used selfing rates to estimate the strength of reproductive isolation between cohorts due to mating system. In CAC_Stream1, maternal families can be divided into three cohorts (*M. guttatus,* admixed, or *M. nasutus*) either according to plot identity or by maternal hybrid index; these classifications agree for 97% of maternal families (all but four: one *M. guttatus* and three *M. nasutus* maternal plants in plots classified as admixed). For simplicity, we chose to use maternal hybrid index as the classifier. For *M. nasutus* (HI>0.95) maternal plants, mating system is an almost complete barrier to reproduction (RI=1.0, Table 3). Between the *M. guttatus* (HI<0.15) and admixed (HI 0.15-0.8) cohorts in CAC_Stream1, mating system was a partial barrier, varying in strength by year and direction (RI=0.026-0.365, Table 3). Classification into cohorts in CAC_Stream2 is more ambiguous, because there is no plot-level division or clearly disjunct distribution between *M. guttatus* and admixed individuals, and because we did not sequence any *M. nasutus* offspring from this stream. If we arbitrarily use the same cutoff as CAC_Stream1 to divide *M. guttatus* and admixed individuals, we find that mating system would provide a similar weak but partial barrier to reproduction between these groups (0-0.113 depending on year and direction, Table 3). Note that all mating system estimates are

**Table 2. Mating system estimation.**

| Cohort[*] | Year | Maternal | | Offspring | | | Selfing |
|---|---|---|---|---|---|---|---|
| | | Families | Fruits | Outcrossed | Selfed | Ambiguous | Rate |
| **CAC_Stream1** | | | | | | | |
| *M. guttatus* (HI<0.15) | 2019 | 21 | 28 | 127 | 20 | 16 | 0.136 |
| | 2021 | 9 | 19 | 133 | 1 | 1 | 0.007 |
| | 2022 | 17 | 18 | 83 | 23 | 6 | 0.217 |
| Admixed (HI 0.15-0.8) | 2019 | 37 | 46 | 225 | 6 | 11 | 0.026 |
| | 2021 | 1 | 3 | 13 | 1 | 8 | 0.071 |
| | 2022 | 15 | 15 | 54 | 31 | 3 | 0.365 |
| *M. nasutus* (HI>0.95) | 2019 | 8 | 8 | 0 | 23 | 2 | 1 |
| | 2021 | 0 | 0 | – | – | – | – |
| | 2022 | 3 | 3 | 0 | 20 | 0 | 1 |
| **CAC_Stream2** | | | | | | | |
| *M. guttatus* (HI<0.15) | 2021 | 4 | 8 | 37 | 0 | 1 | 0 |
| | 2022 | 17 | 17 | 91 | 10 | 2 | 0.099 |
| Admixed (HI 0.15-0.8) | 2021 | 4 | 9 | 42 | 7 | 8 | 0.143 |
| | 2022 | 15 | 15 | 94 | 12 | 6 | 0.113 |

[*]As determined by maternal hybrid index

**Table 3. Measurements of premating reproductive isolation between cohorts.**

| Year | Direction* (ovule ← pollen) | Mating system isolation | Phenological isolation | Combined reproductive isolation† |
|---|---|---|---|---|
| **CAC_Stream1** | | | | |
| 2012 | G ← N | – | 1 | – |
| 2012 | N ← G | – | 1 | – |
| 2012 | G ← A | – | 0.879 | – |
| 2012 | A ← G | – | 0.989 | – |
| 2012 | N ← A | – | -0.51 | – |
| 2012 | A ← N | – | 0.826 | – |
| 2019 | G ← N | 0.136 | 1.000 | 1.000 |
| 2019 | N ← G | 1.000 | 1.000 | 1.000 |
| 2019 | G ← A | 0.136 | 0.917 | 0.928 |
| 2019 | A ← G | 0.026 | 0.886 | 0.889 |
| 2019 | N ← A | 1.000 | 0.591 | 1.000 |
| 2019 | A ← N | 0.026 | 0.684 | 0.692 |
| 2021 | G ← N | 0.008 | – | – |
| 2021 | N ← G | – | – | – |
| 2021 | G ← A | 0.008 | – | – |
| 2021 | A ← G | 0.071 | – | – |
| 2021 | N ← A | – | – | – |
| 2021 | A ← N | 0.071 | – | – |
| 2022 | G ← N | 0.217 | 0.987 | 0.990 |
| 2022 | N ← G | 1.000 | 0.583 | 1.000 |
| 2022 | G ← A | 0.217 | 0.257 | 0.418 |
| 2022 | A ← G | 0.365 | 0.469 | 0.663 |
| 2022 | N ← A | 1.000 | -0.267 | 1.000 |
| 2022 | A ← N | 0.365 | 0.971 | 0.982 |
| **CAC_Stream2** | | | | |
| 2021 | G ← N | 0.000 | – | – |
| 2021 | N ← G | – | – | – |
| 2021 | G ← A | 0.000 | – | – |
| 2021 | A ← G | 0.143 | – | – |
| 2021 | N ← A | – | – | – |
| 2021 | A ← N | 0.143 | – | – |
| 2022 | G ← N | 0.099 | – | – |
| 2022 | N ← G | – | – | – |
| 2022 | G ← A | 0.099 | – | – |
| 2022 | A ← G | 0.113 | – | – |
| 2022 | N ← A | – | -0.054 | – |
| 2022 | A ← N | 0.113 | 0.919 | 0.928 |

Values are scaled from -1 (complete disassortative mating)

to 1 (complete assortative mating), with 0 indicating random mating.

*G=*M. guttatus*, A=Admixed, N=*M. nasutus*.

† Combined effects of phenological and mating system isolation.

Missing values could not be estimated with our dataset.

based on the maternal plant and therefore do not account for paternal effects of selfing on reproductive isolation, such as lower contributions to the outcrossing pollen pool by self-fertilizing flowers.

### Partial phenological isolation between ancestry cohorts

Another potential source of isolation between ancestry cohorts is flowering phenology, so we conducted a multi-year census of flowering phenology at CAC (with plot-level data from 2012, 2019, and 2022; and individual-level data from 2019 and 2022) to address phenological isolation. For a given plot, peak flowering times were typically consistent across years (S7 and S8 Figs), despite differences in overall flower abundance (S7 and S8 Figs). As previously shown for the 2012 growing season [90], we found that plot phenology at CAC consistently tracks hybrid ancestry: median flower date of plots is highly correlated with their median hybrid index (adjusted $r^2 = 0.776$, Fig 4D and S2 Table). The flowering time of individual plants is also associated with ancestry within CAC_Stream1 for both 2019 and 2022 (adjusted $r^2 = 0.435$, Fig 4E and S2 Table): *M. nasutus* flowers were typically marked early in the season, admixed individuals mid-season, and late-season *M. guttatus* individuals later in the spring. In contrast, individual flowering time was not associated with hybrid index for CAC_Stream2 (Fig 4E; see effect of stream and interaction effect of flower date by stream on hybrid index, S2 Table, p=0.0006 and 0.0040), though this may be due to a lack of variation in CAC_Stream2 hybrid index, since we did not mark any CAC_Stream2 *M. nasutus* individuals and the stream lacks a distinct *M. guttatus* cohort.

Using plot-level flowering census data from 2012, 2019, and 2022, we calculated the strength of phenological isolation among the three cohorts in CAC_Stream1: *M. nasutus* (plots S1_3 and S1_6), admixed (plots S1_2 and S1_4), and M. guttatus (plots S1_5 and S1_8); note as before that these plot assignments are generally predictive of individual-level ancestry, but not perfectly so (Fig 4A). Isolation between *M. nasutus* and *M. guttatus* was complete with zero flowering overlap in 2012 and 2019, and strong but incomplete in 2022 (RI=0.58 to 1.0, Table 3). This finding, along with the absence of any first-generation hybrids among 458 wild-collected CAC and LM samples (S3A Fig), suggests new interspecific crosses are rare. In contrast, phenological isolation between admixed individuals and *M. guttatus* was highly variable across the three study years: although the two groups were strongly isolated in 2012 (RI=0.88 and 0.99 depending on direction, Table 3) and in 2019 (RI=0.89 and 0.92, Table 3), they were much less so in 2022 (RI=0.26 and 0.47, Table 3).

What might explain the reduction in phenological isolation in 2022? Although median flowering time of each cohort was relatively stable across years, the duration of flowering was not: admixed plots in 2022 flowered later into the season than in 2019 or 2012 (Figs 4D and S7). As a result, there was greater phenological overlap between cohorts in 2022 compared to 2019 or 2012 (e.g., plots S1_1 and S1_2 overlapping with S1_5; Figs 4D and S7). We see this pattern in the individual-level data as well: in 2019, all marked flowers after May 15 were from the *M. guttatus* cohort, but in 2022, multiple flowers marked after May 15 were from substantially admixed individuals (Fig 4E). Differences in absolute abundance likely play an important role as well: 2019 had much lower flower counts throughout the season in all plots compared to both 2012 and 2022 (S7C and S8C Figs). We also counted relatively more flowers in admixed than in *M. guttatus* plots in both 2012 and 2022 (S7C and S8C Figs), explaining the asymmetry of reproductive isolation between the *M. guttatus* and admixed cohorts in those years (i.e., higher probability of pollen flow from admixed to *M. guttatus* plots: Table 3).

The strength of phenological isolation between admixed individuals and *M. nasutus* also varied across years, depending on the direction of pollen flow. In CAC_Stream1, admixed individuals were less likely to receive pollen from *M. nasutus* in 2022 (RI = 0.97) than in 2012

(RI = 0.83) or 2019 (RI = 0.68), in part due to a wave of late-flowering hybrids in 2022 (Table 3 and Figs 4B and S7B) that did not overlap with *M. nasutus*. For the reverse direction, in both 2012 and 2022, *M. nasutus* was poorly isolated from admixed individuals (RI = -0.51 and -0.267: Table 3), but these estimates are incomplete because our census of open flowers in these years began after *M. nasutus* had already begun flowering (S7B Fig). While CAC_ Stream2 data are more limited, there was substantial phenological isolation preventing pollen flow from *M. nasutus* into admixed plots (RI=0.92, Table 3), but not in the reverse direction, although this asymmetry is due to a higher number of admixed than *M. nasutus* plots, which may not be representative of the stream demography as a whole.

## Measured reproductive barriers accurately predict offspring ancestry shifts

Our paired sets of maternal-offspring genotypes allow us to directly test whether the strong premating barriers we detect at CAC actually translate to observed offspring identities. Any deviation in offspring hybrid indices from maternal values implies incomplete assortative mating (postmating barriers could shift allele frequencies at particular loci but are unlikely to cause systematic shifts in offspring HI). For *M. nasutus*, we observed nearly complete assortative mating due to self-fertilization: at CAC none of our 43 inferred offspring of *M. nasutus* plants across 11 fruits were inferred as outcrossed by BORICE (Table 2). We do find one example at LM where 2 of 7 offspring in a single *M. nasutus* fruit have admixed ancestry (HI=0.687 and 0.627), implying a partially-admixed *M. guttatus*-like pollen parent. In the reciprocal direction, one offspring from a 2022 CAC *M. guttatus* fruit (maternal HI=0.089) and one offspring from a 2021 LM admixed fruit (maternal HI=0.316) had hybrid indices consistent with a *M. nasutus* pollen parent. We can therefore estimate that the rate of *M. nasutus* mating outside its own ancestry cohort is 2/67 or 3.0% maternally, and 2/1501=0.1% paternally.

Next, we asked to what extent mating system differences and phenological isolation shape patterns of assortative mating between hybrids and *M. guttatus* at CAC. We reasoned that assortative mating should be maximized in selfed offspring and, indeed, we observe little to no deviation in HI for offspring inferred as selfed relative to their maternal HI values (Fig 5A and S2 Table). We do find a slight effect of year on this offspring deviation (which may be attributed to uncertainty in hybrid index estimation), but no effect of maternal HI or stream (S2 Table). For outcrossed offspring (which, by definition, have escaped the effects of mating system isolation), we might expect to observe a stronger deviation between maternal and off-spring HIs if assortative mating due to phenological isolation is incomplete. This is precisely what we observe: hybrid indices of outcrossed offspring show a clear deviation from maternal values, with 45.9% of variance in offspring HI deviation attributed to maternal HI (Fig 5B and S2 Table). On average, *M. guttatus* mothers produced more-admixed offspring and admixed mothers produced less-admixed offspring than themselves.

Strikingly, we also find evidence that yearly variation in assortative mating mirrors patterns of phenological isolation at CAC. In 2019, when phenological isolation between *M. guttatus* and hybrids in CAC_Stream1 was nearly complete (RI: 0.92 and 0.88, Table 3), low-HI (*M. guttatus*-like) maternal plants almost never produced offspring with shifts in ancestry. In both 2019 and 2021, most large shifts in hybrid ancestry occurred in the highest-HI admixed indi-viduals (i.e., HI <0.5), which are further from the population mean; they tended to produce offspring with lower hybrid indices, presumably by crossing with more abundant lower-HI hybrids (Fig 5B). In contrast, in 2022, when phenological isolation between these groups was at its lowest (RI: 0.26 to 0.47, Table 3), both low-HI *M. guttatus* and higher-HI admixed parents produced offspring with HIs shifted towards the population mean (Fig 5B), consistent with a breakdown in assortative mating allowing cross-cohort outcrossing. This difference

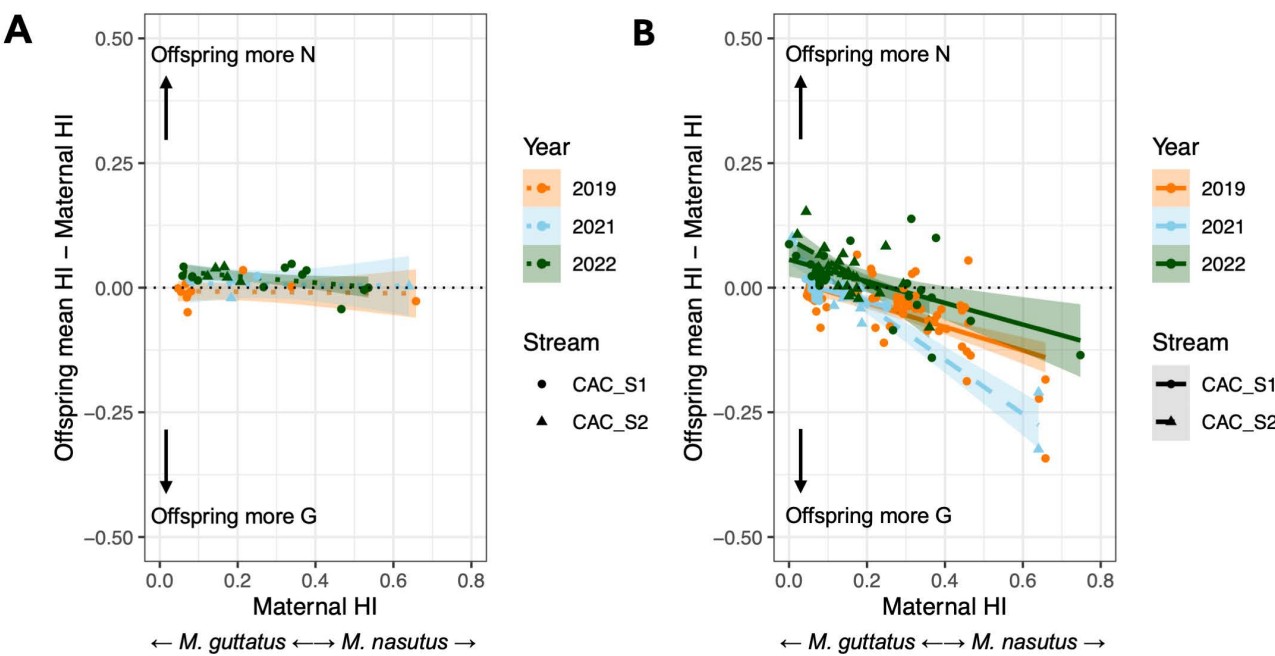

**Fig 5. Assortative mating agrees with patterns of reproductive isolation.** A) Comparison of offspring hybrid index to maternal hybrid index (HI, the proportion of *M. nasutus* ancestry across the genome) for selfed offspring from CAC_Stream1 (CAC_S1) and CAC_Stream2 (CAC_S2). Each point represents the mean of all offspring within a single fruit. Offspring HI deviation is equal to (Offspring HI – Maternal HI), with positive values indicating greater *M. nasutus* ancestry in offspring relative to maternal plants, negative values indicating less *M. nasutus* ancestry, and zero indicating that offspring HI exactly matches maternal HI. Lines indicate individual linear GAM model fits for each year, with shading around each line indicating 95% confidence intervals. There was no significant effect of stream on offspring deviation for selfed offspring (S2 Table), so the two streams are combined into a single linear fit for each year. The thin black dotted line shows the null expectation for selfed fruits, where offspring HI perfectly matches maternal HI. B) Equivalent plot for outcrossed offspring, showing a larger deviation from maternal ancestry, with the direction of deviation typically in the direction of the population mean. Each line indicates a linear GAM model fit for each year and stream, with shading indicating 95% confidence intervals.

is reflected in a significant effect of year (2022) on offspring HI deviation (p=0.01, S2 Table). High-HI individuals were also more likely to self in 2022 (Table 2). The net result of these changes is that, in 2022, outcrossed offspring overall experienced a slight shift to higher levels of admixture compared to maternal samples (mean shift in HI = +0.0172 ± 0.0002, Fig 5B), whereas in 2019 and 2021, there was a slight shift to lower levels of admixture (mean shift in HI = -0.0333 ± 0.0002 and -0.0155 ± 0.0004, Fig 5B).

## Discussion

Here, we describe in detail the composition and mating dynamics within hybridizing populations of *Mimulus guttatus* and *Mimulus nasutus* during secondary contact. Admixture is prevalent in independent streams, with similar patterns of multi-generational, directional introgression from *M. nasutus* into *M. guttatus*. The distribution of hybrid ancestry across space is variable both at very fine (~50m) and coarser (~2.5 km) scales, but stable across a decade of sampling. We measure partial reproductive isolation between three ancestry cohorts by mating system and phenology, both of which likely contribute to the maintenance of ancestry structure across space. We demonstrate substantial year-to-year variation in phenological isolation, which may be associated with climatic variation. Using direct measurements of offspring ancestry composition, we confirm that variation in measured reproductive isolation predicts observed assortative mating. This is a rare direct confirmation of the effect

of premating reproductive barriers on offspring outcomes. Our system demonstrates that the outcomes of hybridization can be dynamic and complex, highlighting the importance of sampling across both space and time.

## Repeated cases of introgression across the landscape

While introgression between *M. nasutus* and *M. guttatus* has been detected previously at Catherine Creek and in a separate sympatric area in California [44,46,89,90,99], strong premating reproductive barriers between the two species imply that initial hybridization rates should be rare [66]. Our data suggest that ongoing admixture is common across the landscape, with independent admixed populations at Little Maui and Catherine Creek, despite no new hybridization events detected. This follows a trend in other systems where genomic signatures of introgression often co-occur with strong reproductive isolation [68,100], and a broader pattern of frequent signals of hybridization despite generally strong premating reproductive isolation across the tree of life [14,20,21]. Theory shows that even occasional migration events can strongly influence allele frequencies [68,101], and rare hybridization events can result in admixed populations if hybrids are persistent once formed. Our finding of weaker reproductive isolation between admixed groups compared to non-admixed progenitors means that, once a few admixed individuals are present, they may promote additional admixture, acting as a genetic 'bridge' between otherwise isolated populations and increasing the chance of adaptive introgression [39,40,102,103]. Admixed groups might also function as a genetic 'sieve': multiple generations of selection can purge incompatible allele combinations, while recombination breaks up linkage between incompatibilities and potentially adaptive alleles [7,46,104].

An open question is how repeatable introgression patterns will be when hybridization occurs multiple times [35,49,104–106]. The overall pattern of directional introgression, with pervasive *M. nasutus* ancestry in majority-*M. guttatus* genomes but very little signature of introgression into *M. nasutus*, is consistent across streams. The high selfing rate of *M. nasutus* explains this directionality, which is a common pattern in selfer-outcrosser pairs [77,99,107–109]. But despite similar asymmetries, each of our three streams have their own unique distribution of hybrid ancestry. This matches findings in other hybrid zones with a mosaic structure [37,104,106,110,111]. Differences in the timing and extent of water availability [77], the spatial distribution of microsites [111], the relative abundance of each species [110], or the presence of pollinators [112] could all influence these idiosyncratic patterns. We note that our data were not designed to capture the relative abundance of *M. nasutus* vs. *M. guttatus* (we likely undersampled *M. nasutus*), but more deliberate sampling in the future could test whether relative abundance impacts differences in the average ancestry of admixed individuals across streams. Expanding this work to additional streams and measuring ecological variables at higher resolution will help us understand the effects of these other factors. Overall, we see that a combination of consistent (i.e., selfing) and heterogeneous (i.e., flowering time) reproductive barriers can produce a patchwork of broadly similar but subtly different outcomes each time admixture occurs.

## Complete organellar capture by hybridization

Hybridization often moves organellar genome haplotypes from one lineage into another, a phenomenon known as organellar capture [113–115]. Typically, organellar capture is assessed in just a small number of samples at phylogenetic resolution. Our population-scale sample provides a unique window into hybridization history – mainly, that an entire sympatric area has a single maternal origin. An *M. nasutus* maternal origin is consistent with the asymmetric nature of gene flow in our system and the observation, by us and others [66] that *M. nasutus*

is more often the maternal parent when hybridizing. It also corroborates our finding that all our CAC and LM *M. guttatus* samples have at least small amounts of nuclear *M. nasutus* ancestry. Still, the extent of capture in even our most *M. guttatus*-like samples is striking and gives us insight into the formation of our hybridizing populations. Not only was the initial hybridization directional, but hybrids must have consistently remained as the seed parent across multiple generations of backcrossing, with nearby *M. guttatus* progenitors contributing primarily through pollen flow rather than seed dispersal. This pattern is consistent with pollen flow acting over longer distances than seed dispersal, so that most new seeds are from maternal plants from within a population, but pollen occasionally arrives from elsewhere. We might expect a similar pattern in other plant systems when pollen flow happens over longer distances than seed dispersal [116], or in animal systems for which males tend to disperse longer distances than females [117].

Another possibility is that the *M. nasutus* organellar haplotype might have some selective advantage within hybrid populations, perhaps due to segregating cytonuclear incompatibilities. Such interactions are common across eukaryotes [118–122], including between populations of *M. guttatus* and *M. nasutus* [123]. But we stress that asymmetries in reproductive isolation and dispersal are sufficient to explain these patterns without needing to invoke selection.

## Persistent spatial structure with fluctuating reproductive isolation

Repeated sampling across years allows us to see that CAC populations have a stable distribution of hybrid ancestry, suggesting a lack of severe hybrid breakdown or maladaptation, though weak postzygotic barriers have been detected [44] and there is genomic evidence of selection against *M. nasutus* ancestry at Catherine Creek [44,90]. Ancestry levels are stably differentiated at ~20-50m scales within one stream (CAC_Stream1), much smaller than the scale of expected pollinator movement, suggesting that forces other than distance are maintaining isolation between cohorts. In contrast, this cohort structure is not apparent in the other streams we sampled (CAC_Stream2 and LM), despite similar distances between plots. The combination of factors that allow multiple ancestry cohorts to persist in one stream (CAC_Stream1) might be idiosyncratic to a particular heterogeneous landscape or population history, and are not necessarily replicable across different populations (e.g., CAC_Stream2 or LM); future work should examine what might be different about the landscape or history of these streams to drive differences in structure.

Population structure at fine scales has been described as a feature of *M. guttatus* populations [124,125], which often harbor substantial genetic diversity; but our study is, to our knowledge, the first to describe hybrid ancestry as a key component of this fine-scaled structure. Mosaic hybrid zones with stark fine-scaled structure have been described in other systems [126,127] and may be common in plants [128] but there are few studies of change over time in such systems. More broadly, hybrid zone studies have sometimes found stability across years [86,87], and other times found substantial shifts [9,84,85], but the reasons for these patterns are underexplored. We therefore set out to understand what components of reproductive isolation exist between cohorts within this fine-scaled mosaic, and how these components of reproductive isolation might change over time. Reproductive barriers between cohorts may serve as one force maintaining a stable mosaic distribution at such fine scales.

As expected, selfing is an important barrier isolating *M. nasutus* from both *M. guttatus* and admixed cohorts. Interestingly, it also provides a partial barrier between *M. guttatus* and admixed cohorts. For *M. guttatus,* our selfing rates generally agree with those of other studies, which range from about 25–50% [124,129–132]. Selfing rates in admixed *Mimulus* have not previously been documented; we find that they are generally closer to *M. guttatus* than *M.*

*nasutus*, consistent with dominance of *M. guttatus* floral phenotypes [95]. We also see that selfing rates fluctuate slightly across years, possibly due to changes in pollinator abundance and timing [133], or in the size and number of flowers as a consequence of general plant health [134]. One intriguing possibility is that the milder conditions in 2022 allowed more plants to have multiple flowers open simultaneously, leading to an increase in geitonogamous (between-flower) selfing, an important selfing mode in *M. guttatus* [130]. But overall, selfing is probably a fairly consistent partial barrier across years.

Phenological isolation is also a strong barrier, confirming previous results [90], and is particularly important between the *M. guttatus* and admixed cohorts of CAC_Stream1, which are less isolated by mating system. However, phenological isolation is quite variable across years, a result confirmed by differential shifts in offspring ancestry across years. Phenology has a strong genetic component in these species: two major QTL contributing to differences in photoperiod response have been mapped to candidate genes [97]. Our results suggest that these genetic differences are only part of the picture, with isolation mediated by other factors. While based on a limited number of years, we note an intriguing trend of stronger isolation in drier (i.e., 2012, 2019) compared to wetter (i.e., 2022) seasons (Fig 1B), pointing to water availability as a potential factor influencing the strength of phenological isolation, acting through changes in flowering duration and abundance. 2022 was also an outlier for lower flowering-season temperatures (Fig 1C), which could play a role as well. Precipitation amount and variability are frequently associated with shifts in flowering phenology across plant taxa [135–138]. With ongoing sampling at Catherine Creek, we will be able to test whether the correlation between precipitation and phenological isolation holds across time. Direct measurements of water availability in microsites across the growing season will also help confirm this relationship in the future.

## Impact of fluctuating environments on hybrid zones

Fluctuating environmental conditions can provide a form of balancing selection, maintaining allelic diversity by favoring different alleles in different years [60,78,80]. Similarly, environmental fluctuations may help maintain a variety of ancestry combinations after admixture: [77] found a correlation between interannual variance in precipitation and the extent of introgression across replicate contact zones. Environmental fluctuations could influence the distribution of hybrid ancestry in two main ways: varying the strength of selection against hybrids [83], or directly modulating the strength of premating reproductive isolation [75,76]. In our case, it appears the environment may be directly impacting the strength of premating reproductive isolation by varying the length of the flowering season and the consequent extent of flowering overlap; however, future work is needed to tie these fluctuations explicitly to particular environmental cues. Reproductive barriers that are sensitive to environmental cues, like phenology, may be important sources of fluctuating reproductive isolation across systems.

Climatic variability is projected to increase in many ecosystems with global climate change [139–141]. It is therefore imperative that we understand the effects of environmental variability on populations. In the context of climate change, hybridization is predicted to have both beneficial effects, such as increased genetic diversity and adaptive potential, and deleterious effects, such as swamping of rare taxa and homogenization of genetic diversity, each of which is likely to be context- and system-specific [75,142–146]. Our data raise the possibility that climatic variability itself can impact the extent of reproductive isolation in hybridizing populations. In a warmer, more unpredictable world, fluctuating reproductive isolation may become more common, leading to an increase in complex, dynamic scenarios of hybridization like this one. To manage these scenarios, we need a better understanding of which factors influence reproductive isolation and hybridization, how they change across space and time, and how they impact the structure and composition of admixed populations.

## Materials and methods

### Environmental data

We obtained daily climatic data for the years 2010–2022 from the PRISM online database [147], dataset ID AN91d, for the 4km grid square centered on (Latitude=45.7113, Longitude=-121.3637), which includes the Catherine Creek site. We calculated a rolling 14-day sum of total daily precipitation and a rolling 14-day mean of daily mean temperature, using the R package 'zoo' [148].

### Field sampling and collections

We sampled tissue for DNA extraction and sequencing from wild-growing *Mimulus* individuals during the 2019, 2021, and 2022 growing seasons (April through June). In 2019, we sampled from a single stream at Catherine Creek, CAC_Stream1. In 2021 and 2022, we sampled from both CAC_Stream1 and an adjacent parallel stream ~120m to the west, CAC_Stream2 (Fig 1A). In 2021, we also sampled from a third more distant stream ~2.5 km away (Little Maui, or LM). In 2021, because we arrived at the field sites in mid-May (due to the ongoing COVID19 pandemic), CAC samples were limited to late-flowering individuals (flowering at LM is shifted to later in the season and thus was unaffected).

Within each stream, we set out 0.5m x 0.5m plots in sites along the stream where *Mimulus* individuals were growing; all samples were collected from within these plots. Plot locations were not always exactly the same across years; to simplify comparisons across years, we assigned plots within 10 m of each other to the same plot ID, resulting in a total of 8 plot IDs each for CAC_Stream1 and LM, plus 7 plot IDs for CAC_Stream2 (Fig 1C). Some plot IDs were not sampled in every year. A few plot IDs encompass samples from more than one adjacent plot from the same year; for these plots, we chose the one with the median number of flowers as a representative for phenological comparisons. A full accounting of plot locations across all years is provided in S4 Table. Approximately once per week throughout the flowering season in both 2019 and 2022, we counted the number of open *Mimulus* flowers within each plot. The total number of flowers across *M. guttatus, M. nasutus,* and potential hybrids was recorded for each plot at each time point; individual flowers were not field-identified to species as hybrids are difficult to distinguish in the field. Flowers are typically open for 1–3 days, and a single individual may have multiple flowers both simultaneously and in sequence (individual flower number ranges from 1 to >50). Phenology data were not collected in 2021.

During each visit to CAC in 2019 and 2022 and LM in 2021, we used acrylic paint to mark the calyx of three random open flowers per plot (if available). We used a different color of paint on each visit to indicate the date flowers were open. Later in the season, we attempted to relocate these same individuals and, if successful, collected fruits from the marked flowers, as well as leaf tissue into envelopes with silica for DNA. Seeds from one to three fruits per maternal plant were later germinated in the UGA Botany greenhouses; leaf/bud tissue was collected from the resulting offspring (1–16 offspring per wild-sampled maternal plant; final counts of families, fruits, and offspring are in S3 Table) and stored at -80°C for DNA extraction and sequencing. Sample metadata is archived at the Dryad Digital Repository [149].

### DNA Extraction and Illumina sequencing

We extracted genomic DNA from both the dried wild-collected tissue samples and the greenhouse-grown offspring samples using a CTAB extraction protocol with phenol-chloroform extraction [150]. Dried tissue was flash-frozen in liquid nitrogen immediately before grinding; fresh tissue from greenhouse-grown offspring was kept at -80C until flash-freezing and grinding.

DNA yield was quantified using a Quant-iT DNA quantification kit (Invitrogen P11496) and plate reader, then normalized to equal concentrations for library preparation.

To prepare libraries for Illumina sequencing, we used a low-coverage Tagmentation approach [151,152]; our protocol is available at [153]. Briefly, Tn5 enzyme was purified in bulk and pre-loaded with universal Illumina adapters following [154]. We then added the loaded Tn5 to approximately 1 ng of sample DNA and incubated to fragment DNA and add universal adapters. Next, we added OneTaq Hot Start polymerase (New England Biolabs M0488L) along with combinatorially barcoded forward and reverse primers and ran 18 cycles of PCR to amplify fragments. After PCR, samples were pooled into sets of 48 and cleaned using SPRI magnetic beads. These sets of 48 samples were quantified with a Qubit fluorometer and then merged in equimolar amounts for Illumina sequencing. Samples were sequenced at the Duke University Center for Genomic and Computational Biology using an Illumina NovaSeq 6000 machine to generate paired-end 150 bp reads with a targeted depth of ~1X coverage per sample. Reads were demultiplexed based on their combinatorial barcodes into individual samples.

Samples were prepared and sequenced across three separate runs: run 1 included a smaller test set of 31 maternal samples from 2019; run 2 included the remaining 2019 maternal and offspring samples (including re-sequencing the original 31) plus all 2021 maternal and offspring samples, and run 3 included all 2022 maternal and offspring samples. After a preliminary analysis on the 31 original samples, with data from runs 1 and 2 treated separately, one sample was excluded because hybrid index values (see *Assigning local ancestry*) obtained from the two runs did not agree; for the remaining 30 twice-sequenced samples, we concatenated their raw read files across both runs for the final analysis. From our full dataset, we also excluded 80 offspring samples whose maternal plant was not sequenced, and 1 maternal family (mother and offspring) with missing location data, leaving a total of 2946 samples. Across these samples, we sequenced ~4.8 billion read pairs, for an average of 1.64 million read pairs per sample (standard deviation 1.57 million read pairs). After filtering to remove samples with less than 25,000 called ancestry-informative sites (see *Assigning local ancestry*), our dataset included 2706 samples with mean sequencing depth of 1.78 million read pairs per sample (standard deviation 1.56 million read pairs, range 45,098 read pairs to 15.72 million read pairs). This final dataset included 458 wild-collected maternal samples and 2248 greenhouse-grown offspring. Details about this final dataset are provided in Table 1.

## Reference alignment, genotyping, and creation of SNP panels

For each Illumina sample, we used Trimmomatic v0.39 [155] to remove adapter sequences and low-quality ends. We aligned reads with bwa v0.7.17 'mem' [156] to the *Mimulus guttatus* IM62 v3 reference genome (https://phytozome-next.jgi.doe.gov), removed duplicates with picard v2.21.6 'MarkDuplicates' [157], and used samtools v1.10 [158] to keep only properly paired reads with map quality>=29. Coverage stats were obtained with qualimap v2.2.1 [159]. Across all 2946 samples, 83% of reads mapped successfully, for an average per-base genome coverage of 0.44 and an average of 14.0% of the genome covered by at least one read. For the 2706 samples retained after filtering (see below), average per-base genome coverage was 0.48 and an average of 15.3% of the genome was covered by at least one read.

We created panels of high-quality, informative SNPs using 36 previously sequenced lines (S5 Table), including one *M. nasutus* and 8 admixed *M. guttatus* lines from CAC, plus 3 additional *M. nasutus* and 24 allopatric *M. guttatus* from throughout the species' ranges. *M. nasutus* has lower range-wide nucleotide diversity ($\pi_{syn}$~1.1%) than *M. guttatus* ($\pi_{syn}$~4.9%) so the smaller *M. nasutus* sample is informative [89]. We followed the same steps above to align these Illumina panel lines to the *Mimulus guttatus* IM62 v3 reference genome. The panel was

genotyped using GATK 4.4.0.0 HaplotypeCaller and GenotypeGVCFs in all-sites mode [160]. Called sites were split into biallelic SNPs and invariant sites, with indels and multiallelic sites removed. SNPs were further filtered with GATK to remove sites that failed a set of standard quality filters (details in the associated code at github.com/mfarnitano/CAC_popgen). Sites were further filtered at the individual genotype level using vcftools v0.1.16 [161], setting genotypes to missing if individual sequencing depth was below 6 or above 100, or if genotype quality was less than 15 for that sample. Heterozygous calls were retained. From the resulting genotype file, we created a list of 3,493,514 SNPs called in at least 31 of the 36 reference individuals.

For a more targeted genotyping panel, we created a test set of 100 representative wild CAC+LM samples, then subset our variant list to 19,633 sites with nonzero read coverage in at least 60% of test samples and a minor allele frequency of at least 20%. This panel should include variants that are segregating in our focal populations, including both species-diagnostic and intraspecific segregating sites, and is tailored for relatively high coverage in our otherwise low-coverage dataset.

In addition, we created a panel of ancestry-informative sites distinguishing *M. guttatus* and *M. nasutus,* subsetting our full variant list to 208,560 SNPs with>=80% allele frequency difference between the 24 high-coverage allopatric *M. guttatus* lines and the four high-coverage *M. nasutus* lines. This panel should include only variants that are species-informative, but includes sites that may have low coverage across our full dataset. Given the high genetic diversity of *M. guttatus* relative to *M. nasutus* and the fact that *M. nasutus* variation is nested within range-wide *M. guttatus* variation [89], any one variant is not species-diagnostic and may be shared through ancestral polymorphism; however, the combination of multiple informative variants across a region, especially at high LD within *M. nasutus* ancestry blocks*,* can be diagnostic for local species ancestry [89,162].

## Population structure analyses with ANGSD

We used ANGSD v0.940 [163] with the GATK likelihood model to obtain genotype likelihoods for all wild-collected samples at the targeted variant list of 19,633 SNPs. We then used these genotype likelihoods to run a genomic PCA analysis with PCAngsd [164]. The resulting covariance matrix was converted into principal components using the eigen function in R [165]. As a complementary approach to our ancestry HMM (below), we used NGSadmix [166] with K=2 groups to estimate admixture proportions using the ANGSD genotype likelihoods.

## Assigning local ancestry

We used ancestry_HMM [162] to assign *M. guttatus* vs. *M. nasutus* ancestry across the genome for each maternal and offspring individual, following the *ancestryinfer* pipeline from [167] and using our 208,560 ancestry-informative sites (above). Reads for each maternal and offspring sample were aligned to both the *Mimulus guttatus* IM62 v3 reference and the *Mimulus nasutus* SF v2 reference (https://phytozome-next.jgi.doe.gov), keeping only reads that aligned exactly once to each genome. At each ancestry-informative site, we counted the number of reads supporting each allele. These read counts per allele were used as input for ancestry_HMM [162], along with allele frequencies in the 24 allopatric *M. guttatus* and 4 *M. nasutus* reference lines. Recombination rates between each SNP position were approximated using the bp distance between adjacent sites multiplied by the global recombination rate estimate of 3.9e-8 Morgans/bp, calculated using a total genetic map length of 14.7 Morgans [89] across a genome size of 375 Mb, approximately the mean genome size between *M. guttatus* (~430 Mb) and *M. nasutus* (~320 Mb). Ancestry_HMM was run with a simple model of two

source populations and a single admixture pulse (specific parameters available within the code at github.com/mfarnitano/CAC_popgen) to estimate posterior probabilities of *M. guttatus, M. nasutus,* or heterozygous ancestry at each site for each sample. Posterior probabilities of at least 0.9 for any genotype were converted to hard genotype calls, with lower values set to missing. 240 samples with fewer than 25,000 called genotypes (out of 208,560 sites) were excluded from all analyses, leaving a final dataset of 2706 samples from an initial set of 2946 samples (Table 1). Hybrid index (HI) equals the number of sites with homozygous *M. nasutus* calls plus half the number of heterozygous calls, divided by the total number of called sites. Ancestry heterozygosity (AH) equals the number of heterozygous calls divided by the total number of called sites.

To test the reliability of our local ancestry measurements, we subset the raw reads from each allopatric *M. guttatus, M. nasutus*, and high-coverage CAC line (36 lines total) to 2 million read pairs per sample, and followed the above local ancestry pipeline to call ancestry across the genomes in these lines. All 4 *M. nasutus* lines had hybrid index >0.99; 18 of 24 allopatric *M. guttatus* lines had HI<0.01 and the remaining six had HI<0.05. The 8 high-coverage CAC *M. guttatus* were more admixed, with hybrid index ranging from 0.11 to 0.34, consistent with previous ancestry estimates for these lines [44].

Our data include 11 maternal samples from plots S2_1 and S2_2, all from 2022, that we determined to be the allopolyploid species *M. sookensis.* These individuals had hybrid indices near 0.5 and ancestry heterozygosity >0.85, which could indicate polyploidy or new F1 hybridization. Offspring of these individuals had very similar hybrid indices and similarly high ancestry heterozygosity, which is consistent with fixed heterozygosity in these highly selfing polyploids [98,168,169] but would not be true for the F2 progeny of F1 hybrids. In addition, these individuals formed a distinct cluster in PCA space: they diverged from the 1–1 correlation between hybrid index and PC1, and they were separated cleanly by PC5. Finally, offspring grown in the greenhouse had a small flower size and shape that matched *M. sookensis* and *M. nasutus;* F1 hybrids typically have larger flowers more similar to *M. guttatus* [95]. We removed all *M. sookensis* individuals from the analyses of reproductive isolation, individual phenology, mating system, and offspring ancestry deviations.

We combined our data with previously obtained hybrid indices from 75 individuals collected in 2012 from CAC_Stream1 and CAC_Stream2 [90]. These samples were genotyped using MSG sequencing [170], and hybrid indices were calculated from an HMM approach implemented in HapMix [171]; note that these are different data types and were processed differently than our 2019–2022 data. However, as both approaches use low-coverage whole-genome sequencing and HMM ancestry calling, we expect them to produce broadly comparable hybrid indices even if there are slight differences at individual loci.

## Organellar haplotype networks

To investigate the haplotype structure of maternally transmitted organellar genomes in our population, we aligned each of our maternal samples, along with 8 high-coverage CAC lines (S5 Table), to the *M. guttatus* IM767 v1.1 chloroplast assembly (https://phytozome-next.jgi.doe.gov) using BWA v0.7.17 [156], and called genotypes using GATK v4.4.0.0 HaplotypeCaller and GenotypeGVCFs functions [160]. Read coverage aligned to the chloroplast genome was high compared to the nuclear genome, so we restricted each analysis to samples with mean coverage>=40, resulting in 328 of our samples and 6 of the previously sequenced CAC lines. We then subset each VCF to a set of SNPs from a recent analysis of organellar haplotypes across *M. guttatus, M.nasutus, M. decorus*, and *M. sookensis* [98], further removing any SNPs with missing data, heterozygous calls, or individual depth <4 for any sample using BCFtools v1.15.1 [158]. We merged this dataset with the genotype calls for the 41 samples

from [98], output SNP calls as a fasta sequence for each sample using BCFtools consensus, and converted to nexus format using EMBOSS v6.6.0 [172]. We then used the Integer NJ Net function in popart [173] to generate a haplotype network from the resulting alignment. Our final chloroplast dataset included 102 segregating sites, of which 43 were parsimony-informative. We followed this same approach to generate a mitochondrial genome haplotype network, aligning sequences to an *M. guttatus* IM62 mitochondrial assembly [174] and excluding regions annotated as chloroplast-derived. After filtering for mean coverage>=40, we retained 150 of our samples and 8 previously sequenced CAC lines, which were merged with the 41 samples used in the recent analysis. Our final mitochondrial dataset included 120 segregating sites, of which 39 were parsimony-informative.

### Estimating selfing rates

To estimate selfing and outcrossing rates, we started with 310 maternal families containing 1,993 offspring where both the maternal sample and at least one offspring sample had at least 25,000 sites with ancestry assignments (see *Assigning local ancestry* above). Then, we excluded maternal-offspring pairs with a large number of genotypes that contradict a maternal-offspring relationship, suggestive of high ancestry error rates in either the maternal or offspring sample, as follows. For each maternal-offspring pair, we used ancestry calls from ancestry_HMM (at 208,560 sites) to calculate the number of informative sites where genotypes were incompatible with maternal-offspring relationships (i.e., maternal and offspring samples were homozygous for opposite ancestries), divided by the total number of homozygous maternal calls with a called genotype in the offspring. Allowing for some uncertainty in ancestry calling, we excluded any offspring for which this incompatibility ratio was>=5%, as well as entire families if greater than half the offspring were excluded. After these filters, our final family dataset contained 185 maternal plants (60% of the original dataset) with offspring, plus 1565 offspring (79% of the original dataset). We note that this filter is designed to remove maternal or offspring samples with poor genotyping quality (high error rates), since selfing rate estimation is sensitive to such errors. The filter is purposefully strict: because our initial ancestry assignments use a posterior probability cutoff of 90%, for any pair of samples, a given site may have up to a 0.1+(0.9*0.1)=19% chance that at least one of the two samples has an incorrect ancestry assignment, prior to this filter. As such, failing this maternal-offspring filter does not necessarily imply that the offspring was incorrectly assigned to the mother, though the filter should remove any such cases as well.

We then randomly thinned our list of ancestry-informative sites to one per 10kb, leaving 17,514 sites, and calculated genotype likelihoods at each site for both maternal and offspring samples using ANGSD v0.940 with the GATK likelihood model [163]. Genotype likelihoods estimated in this way can be biased towards homozygosity when using low-coverage data, because only a single allele is typically sampled at each site. Furthermore, when short fragments are sequenced, forward and reverse paired-end reads can overlap, and ANGSD genotype likelihoods count this single fragment as two separate reads, artificially inflating the likelihood of a homozygous genotype. Since the estimation of selfing rates is sensitive to the likelihood of homozygous vs. heterozygous calls, we chose to mitigate this effect by using alignments of the forward reads only, discarding reverse reads, in our genotype likelihood estimation for selfing rate analyses. We estimated probabilities of selfing vs. outcrossing for each maternal-offspring pair, and probabilities of half-sibling vs. full-sibling relationships for each outcrossed sibling pair, using the Bayesian model implemented in BORICE.genomic v3 [124]. We tuned allele frequencies for 20 steps, then had 20 steps of burn-in and a total chain length of 100, with records kept every 2 steps. Offspring with a posterior probability>= 0.9 were called as selfed or outcrossed, and other offspring were excluded. We counted pairs as

half-siblings if the posterior probability was >50%, full-siblings if it was <50%, and missing data if the posterior probability was exactly 50%.

## Calculating the strength of reproductive isolation

We calculated standardized estimates [175] of reproductive isolation (RI) between three ancestry cohorts within CAC_Stream1, defined by hybrid index: an *M. guttatus* (lower admixture levels <0.15) cohort, an admixed (higher admixture levels >0.15), and an *M. nasutus* cohort. We calculated two components of premating reproductive isolation: mating system isolation, the degree to which self-fertilization reduces cross-cohort mating; and phenological isolation, the degree to which phenological non-overlap reduces cross-cohort mating; we also calculated their combined effects. Each measure ranges from -1 (complete disassortative mating) to 1 (complete assortative mating), with 0 indicating random mating. For CAC_Stream2, we calculated mating system isolation between *M. guttatus* and admixed cohorts only (due to a lack of *M. nasutus* sampling), using the same hybrid index cutoff as CAC_Stream1, though this cutoff may not be meaningful for this stream given its lack of cohort structure. Phenological RI was not calculated for LM due to a lack of the relevant cohort or plot structure; also, LM was only sampled in 2021, a year in which we did not collect phenological data. We did not calculate mating system isolation for LM since almost all (33/35 or 94%) available maternal families after filtering fell into the same admixed hybrid index cohort.

For mating system isolation, we used selfing rates estimated from BORICE, dividing individuals into three cohorts by hybrid index. We assumed for this calculation that outcrossing events contribute no isolation to the maternal cohort (RI=0), while selfing events contribute perfect isolation to their maternal cohort (RI=1). The degree of mating system isolation is then equal to the rate of selfing of maternal plants in each cohort. While selfing flowers may also contribute less to the pollen pool, causing increased mating system isolation from the paternal direction, this indirect effect is not included in our calculation.

To calculate phenological isolation between cohorts within streams CAC_Stream1 and CAC_Stream2, we utilized plot-level phenological data, assigning each plot to an ancestry cohort according to median hybrid index. For CAC_Stream1, we excluded two plots (S1_1 and S1_7) with apparently bimodal hybrid indices indicating the presence of multiple cohorts. CAC_Stream1 plots were assigned as follows: S1_5 and S1_8 representing *M. guttatus*, S1_2 and S1_4 representing admixed, and S1_3 and S1_6 representing *M. nasutus*. For CAC_Stream2, we assigned plots to two cohorts, *M. nasutus* (plots S2_1 and S2_2) and admixed (plots S2_3 to S2_7). For CAC_Stream2, we did not calculate phenology in 2012 because the only sampled *M. nasutus* plot in that year (S2_2) also had a substantial number of admixed individuals. For each pair of cohorts in a stream, we calculated the probability of within-cohort vs. between-cohort matings based on the number of open flowers in each plot group on each day: for each open flower, between-cohort mating probability is equal to B/(A+B) and within-plot mating probability is equal to A/(A+B), where A and B are the number of open flowers on the same day as the focal flower from the same and opposing plots, respectively. This assumes that each open flower contributes equally to the pollen pool and ignores the effects of physical distance. These probabilities were then summed across all open flowers across the season to obtain an estimated total number of between-cohort and within-cohort potential matings. Reproductive isolation was then calculated as RI=1-2P, where P is the relative probability of these between-cohort vs. within-cohort potential matings.

Following [175], we calculated the combined effect of mating system ($RI_M$) and phenological ($RI_P$) isolation using Equation 1 below, for any comparison where we had both values. Note that the order of isolating barriers does not matter for this calculation.

### Equation 1. combined reproductive isolation

$$1. \quad RI_{Combined} = RI_M + \left(\left(1 - RI_M\right) * RI_P\right)$$

### Using offspring to assess assortative mating

If assortative mating by ancestry is strong, we expect offspring hybrid index to closely match maternal hybrid index; in contrast, deviations between offspring and maternal hybrid index may indicate mating across different ancestry cohorts. To test for deviations from maternal hybrid index within the offspring, we first separated each fruit into selfed and outcrossed offspring based on BORICE. We then calculated the mean for each fruit of the difference between Offspring HI and Maternal HI (Offspring HI deviation), averaging selfed and outcrossed offspring separately. We expect a deviation near 0 for selfed offspring, or under strong assortative mating by ancestry; positive values indicate that the offspring have more *M. nasutus* ancestry than their maternal parent, while negative values indicate they have more *M. guttatus* ancestry than their maternal parent.

### Statistical models

All statistical test were carried out using generalized additive models (GAMs) implemented with the R packages 'gam' [176] and 'mgcv' [177]. For each model below, we sequentially added independent variables in the order listed, starting with an intercept-only model and allowing for interactions, and retained a variable only if the variable had a significant main or interaction effect and it significantly improved the model fit according to a likelihood ratio test. All model details are provided in S2 Table. GAMs were used because they could handle a variety of model types (beta, binomial, and linear) and allow for incomplete factor interactions (e.g., not all combinations of stream and year were present in the data).

To test for predictors of selfing rates, we ran binomial GAMs (family = binomial, link = logit) with selfing rate as the dependent variable and the following independent variables: maternal hybrid index, stream (CAC_Stream1 or CAC_Stream2), and year (2019, 2021, or 2022). We ran these models for all families, then for all families except *M. nasutus* (HI>0.95) and for only families with maternal HI<0.5 (i.e., excluding the six majority-*M. nasutus* admixed individuals).

To test whether phenology was associated with hybrid index at the plot level, we paired flower censuses throughout the growing season from 2012, 2019, and 2022 with hybrid indices from our sequenced maternal plants from those plots. For each plot in each year, we calculated the date that the median censused flower was open in that plot. We then took the median hybrid index of all sequenced individuals from that plot. We ran beta regression GAMs (family = betar, link = logit) to test for an effect of median flower date, stream, and year on median hybrid index.

We also tested whether flower date was associated with hybrid index at the individual level. For sequenced individuals in 2019 and 2022, we have dates that marked flowers were open – note that these are not necessarily the first flowers on a given individual but were instead randomly chosen flowers from the plot on the day of marking. We ran beta regression GAMs (family = betar, link=logit) to test for an effect of individual marked flower date, stream, and year on maternal hybrid index.

We ran linear GAMs (family = gaussian, link = identity) to test for an effect of maternal hybrid index, stream, and year on offspring HI deviation (the difference between offspring hybrid index and maternal hybrid index, averaged per fruit). We ran separate linear models

for the selfed offspring and for the outcrossed offspring, as well as a combined model with 'mate type' (selfed or outcrossed) as an additional predictor, added before stream and year. We also ran an ANOVA on each linear model using the 'anova' function in the R package 'stats' [165] to assess the relative effects of each predictor and interaction variable.

## Supporting information

**S1 Fig. Correlation between hybrid index and NGSAdmix structure results.** NGSAdmix value is the proportion assignment to one of two clusters by NGSAdmix with K=2; hybrid index is the proportion of sites with M. nasutus ancestry from local ancestry inference using Ancestry_HMM. The circled 'SOOK' cluster that deviates from the 1–1 line is composed of *M. sookensis* polyploid individuals (see Materials and Methods). 'NAS' = *M. nasutus.* (TIFF)

**S2 Fig. Ancestry heterozygosity vs. hybrid index for maternal and offspring plants.** Ancestry_HMM hybrid index and ancestry heterozygosity outputs for A) maternal and B) offspring individuals. Hybrid index or HI is the proportion of ancestry-informative sites called with *M. nasutus* ancestry from Ancestry_HMM. Ancestry heterozygosity or AH is the proportion of ancestry-informative sites called has heterozygous (out of the total number of called ancestry-informative sites). First-generation hybrids between 100% *M. guttatus* and 100% *M. nasutus* would have an expected HI=0.5 and AH=1.0; their offspring would be expected to have lower ancestry heterozygosity (~0.5 if selfed). The circled 'SOOK' cluster highlights a group of polyploid *M. sookensis* maternal plants; their offspring have similarly high AH values, which indicates fixed heterozygosity (polyploidy) rather than diploid F1 status. (TIFF)

**S3 Fig. PC5 identification of *M. sookensis*.** Principal components 1 and 5 from PCAngsd analysis (from the same analysis as PCs 1–4 shown in Fig 2A and 2B). PC1 correlates strongly with *M. guttatus* vs. *M. nasutus* ancestry (Fig 2B). PC5 clearly delineates a group of individuals identified as the polyploid species *M. sookensis.* (TIFF)

**S4 Fig. PCA analysis of only M. guttatus and admixed individuals from CAC_Stream1 and CAC_Stream2.** A) PC axis 1 vs. hybrid index, demonstrating that ancestry still drives the major axis of genomic variation in the dataset even without *M. nasutus* samples. B) PC axes 1 and 2, demonstrating some separation of certain CAC_Stream1 plots (particularly S1_1 and S1_2) from the rest of the data. C) PC axes 3 and 4, showing further variation within CAC_Stream1 that is partitioned by plot identity. Note that CAC_Stream2 is not differentiated from CAC_Stream1 in these first four axes, but the major axes of variation instead separate variation within CAC_Stream1. LM samples, *M. nasutus* samples, and *M. sookensis* samples were excluded prior to running thi PCA. (TIFF)

**S5 Fig. Complete mitochondrial capture of a *M. nasutus* haplotype in sympatric *M. guttatus* .** NJ-Net haplotype network of mitochondrial variation built from 154 CAC and LM maternal samples sequenced in this study (bolded captions) and 49 previously sequenced samples (non-bolded captions) from the *M. guttatus* species complex [98], using 120 total variant sites (39 parsimony-informative). The mitochondrial network largely agrees with the chloroplast network (Fig 3), with a single haplotype present in all maternal samples from Catherine Creek and Little Maui, including *M. guttatus,* admixed, *M. nasutus,* and *M. sookensis* samples.

All other *M. nasutus* samples from across the range share this haplotype or a close derivative, as do samples from *M. sookensis* (a polyploid with *M. nasutus* as maternal parent). *M. guttatus* haplotypes are more variable, with only one sample not from CAC or LM sharing the *M. nasutus* haplotype. *M. decorus* is another member of the *M. guttatus* species complex with variable mitochondrial haplotypes.
(TIFF)

**S6 Fig. Selfing rate vs. hybrid index.** Proportion of selfed vs. outcrossed offspring per maternal family vs. maternal hybrid index (proportion *M. nasutus* ancestry of the maternal plant), showing (A) a binomial model fit including *M. nasutus* maternal families, and (B) a binomial model fit excluding *M. nasutus* families. Self vs. outcross determined by the BORICE Bayesian model. Offspring without>=90% posterior probability of either state were removed. Lines and shaded regions indicate model fits and 95% confidence intervals from a binomial GAM regression using stream and year as additional factor variables (S2 Table).
(TIFF)

**S7 Fig. Comparison of hybrid index, phenology, and maximum floral count for each plot within CAC_Stream1.** A) Histograms of binned hybrid index for sampled maternal plants in each CAC_Stream1 plot, with years as stacked colors. 0=*M. guttatus,* 1=*M. nasutus.* B) Phenology (counted open flowers) throughout the flowering season in seven plots within CAC_Stream1 during the 2012, 2019, and 2022 seasons, scaled to the maximum number of open flowers counted within that plot in that year. Plots with primarily *M. nasutus* individuals (plots S1_1,S1_3, and S1_6) had early flowers, plots with more admixed individuals (plots S1_2, S1_4, and S1_7) had intermediate peak flowering, and plots with primarily *M. guttatus* (plots S1_5 and S1_8) had later peak flowering. Peak flowering was similar but not identical across years. Tails of open flowers indicate that in 2022, plots with admixed individuals (plots S1_1,S1_2, and S1_4) continued flowering for much longer than in 2019, resulting in more overlap with *M. guttatus* plots (plots S1_5 and S1_8). C) Maximum number of open flowers on any one day for each plot. In 2019, there were far fewer open flowers throughout the growing season than in 2021 or 2022.
(TIFF)

**S8 Fig. Comparison of hybrid index, phenology, and maximum floral count for each plot within CAC_Stream2.** D) Histograms of binned hybrid index for CAC_Stream2 plots. E) Phenology (counted open flowers) for CAC_Stream2 plots, scaled to the maximum number of open flowers counted within that plot in that year. F) Maximum number of open flowers for CAC_Stream2 plots.
(TIFF)

**S1 Table. Distribution of samples from each plot in each PCA cluster.**
(XLSX)

**S2 Table. Statistical model output summaries.**
(XLSX)

**S3 Table. Summary of selfing and sibship estimation from BORICE.**
(XLSX)

**S4 Table. Plot names, locations and ID assignments for each year of sampling.**
(XLSX)

**S5 Table. Reference panel lines used for SNP panel creation.**
(XLSX)

## Acknowledgments

We thank Alex Sotola, Sara Hill, Autumn Knight, Liza Lesley, and Sherwin Shirazi for help processing samples, and Bob Schmitz for providing tagmentation enzyme for library preps. Thank you to Alex Sotola, Jill Anderson, Molly Schumer, Casey Bergman, and Kelly Dyer for advice during project development. We also thank Eleanore Ritter, Natalie Gonzalez, Logan Scott, and Sam Mantel for comments on an earlier draft of the manuscript.

## Author contributions

**Conceptualization:** Keith Karoly, Andrea L Sweigart.

**Data curation:** Matthew C Farnitano, Keith Karoly, Andrea L Sweigart.

**Formal analysis:** Matthew C Farnitano.

**Investigation:** Matthew C Farnitano.

**Methodology:** Matthew C Farnitano, Keith Karoly, Andrea L Sweigart.

**Project administration:** Andrea L Sweigart.

**Resources:** Andrea L Sweigart.

**Supervision:** Andrea L Sweigart.

**Validation:** Matthew C Farnitano.

**Visualization:** Matthew C Farnitano.

**Writing – original draft:** Matthew C Farnitano.

**Writing – review & editing:** Matthew C Farnitano, Keith Karoly, Andrea L Sweigart.

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
