## [Decision Letter · Decision Letter 0]

8 Nov 2024

PGENETICS-D-24-01091Fluctuating reproductive isolation and stable ancestry structure in a fine-scaled mosaic of hybridizing *Mimulus*  monkeyflowersPLOS Genetics Dear Dr. Farnitano, Thank you for submitting your manuscript to PLOS Genetics. After careful consideration, we feel that it has merit but does not fully meet PLOS Genetics's publication criteria as it currently stands. Therefore, we invite you to submit a revised version of the manuscript that addresses the points raised during the review process. Please submit your revised manuscript within 60 days Jan 07 2025 11:59PM. If you will need more time than this to complete your revisions, please reply to this message or contact the journal office at plosgenetics@plos.org. Please include the following items when submitting your revised manuscript:* A rebuttal letter that responds to each point raised by the editor and reviewer(s). You should upload this letter as a separate file labeled 'Response to Reviewers '. This file does not need to include responses to any formatting updates and technical items listed in the 'Journal Requirements' section below.* A marked-up copy of your manuscript that highlights changes made to the original version. You should upload this as a separate file labeled 'Revised Manuscript with Track Changes '.* An unmarked version of your revised paper without tracked changes. You should upload this as a separate file labeled 'Manuscript '. If you would like to make changes to your financial disclosure, competing interests statement, or data availability statement, please make these updates within the submission form at the time of resubmission. Guidelines for resubmitting your figure files are available below the reviewer comments at the end of this letter. We look forward to receiving your revised manuscript. Kind regards, Jesse LaskyAcademic EditorPLOS Genetics Justin FaySection EditorPLOS Genetics

Aimée Dudley

Editor-in-Chief

PLOS Genetics

Anne Goriely

Editor-in-Chief

PLOS Genetics

**Journal Requirements:** **Additional Editor Comments (if provided):** I believe the reviewers have made many helpful suggestions. In particular I think it is important to address concerns over sub-setting the samples to particular ('representative') years and locations: how those decisions were made, why subset at all. I also agree that conclusions about microhabitat variation based on the data here do not seem supported.**Reviewers' comments:** Reviewer's Responses to Questions

**Comments to the Authors:**

Reviewer #1: 1. Fig.2 shows that CAC_A and CAC_B only contain samples from CAC_S1 stream (open circles) and CSC_C contains both. However, it seems CAC_C tends to have more samples of year 2021 and 2022 while the others tend to have samples of year 2019? And it is not easy to understand “genetic differentiation between samples separated by as few as 50 m” (Line 170) as CAC_S1 and CAC_S2 still shared the same cluster CAC_C in Fig.2 So it would be good to label the three distinct clusters of admixed individuals (microspatial structures) in the map of Fig. 1C. It is quite surprising to observe genetic differentiation within 50 m for the outcrosser M. guttatus. Is there any possibility of severe inbreeding of M. guttatus within a short sampling distance? Please highlight two species in Fig.2 A lower panel.

2. How many M. sookensis in this study and how many are allopolyploid? Line 183 says 11 from CAC_S2 with His ~0.5 were allopolyploid species M. sookensis and were removed from further analyses. I found 16 M. sookensis (9 from CAC) in Fig3 major haplotype. Table S1 has 11.

3. Line 197 – 200 “In contrast, the haplotype is almost never found in M. guttatus, which has much higher organellar diversity … only two carry this CAC/LM haplotype”. I am not quite sure I understand this expression. I guess you mean the major haplotype is carried by all M. nasutus and M. sookensis but is rare in non-CAC M. guttatus with only two accessions (BOG10, DPR84). How about the 41 LM M.guttatus + admixed?

4. Line 202 – 206 “We therefore infer that this haplotype is derived from M. nasutus.. is free from the effects of introgression”. I am not sure one can reach at these conclusions simply based on comparing the number of individuals carrying haplotypes in both species, but neither considering the effects of genetic drift nor selection on maintenance of introgressed alleles. For example, Ne could be much smaller in M. nasutus as it is selfing and thus introgressed allele can be fixed or lost at a much faster speed than the outcrosser.

5. Line 213 “HI=0.1-0.2 for LM and 0.23-0.3 for CAC_S2” Is there any possibility of switching the values by mistake?

6. Line 228 “plot S1_5B, for example, … ” Is that simple because sample size is too small in year 2012 in this plot?

7 Line 417-421. The authors used all evidence to show the mating system and phenology difference caused asymmetric gene flow and I think that is a very neat explanation. On the other hand, no evidence in this study showed better adaptation of M. nasutus. One needs to explain more why cytonuclear incompatibility grants disadvantage to M. guttatus and admixed if the possibility of selection needs to be discussed. Comparing to selection, I think drift with small and large Ne in selfer and outcrosser could be more interesting.

8. It seems only precipitation data has been collected and compared. How about other environmental variables such as max or min temperature which is expected to affect phenology or the pollinators as well.

9 How many genetic loci were amplified? Are they random? What is the proportion of genome covered?

10 The discovery panel is quite small, especially only four M. nasutus were sequenced, for allele frequency estimation. How does one guarantee there is no ancestral polymorphism exists even in allopatric sites? What is the average genetic distance (FST or Dxy) for two species? I think one at least should discuss the possibly consequences of mis-assignment of ancestry.

Reviewer #2: Review of “Fluctuating reproductive isolation and stable ancestry structure in a fine-scaled mosaic of hybridizing Mimulus monkeyflowers”

Summary of research:

This study investigates hybrid population structure and reproductive isolation between M. guttatus and M. nasutus. The authors combine genetic data from wild individuals and greenhouse-reared offspring with phenological data to explore how phenology and mating systems influence introgression dynamics across years. The study contributes to the growing literature documenting various outcomes of hybridization. They empirically detail how variation in phenological isolation of M. guttatus and M. nasutus across years corresponds with admixture frequency. The study also quantifies how differences in mating system (selfing vs. outcrossing) result in strongly directional gene flow - with the selfer admixing much less frequently, as predicted by theory. They find that ancestry proportions generally remained stable in studied populations.

While none of the claims are particularly novel, the work provides uncommon empirical validation for commonly discussed theory (eg that phenological isolation impacts hybridization rates). The sequencing of maternal-offspring pairs is a particularly nice component of the study.

Major comments:

In general, the naming and use of different sites is difficult to follow (For example, why are there only a few plots with A and B?). Consider ways to simplify/clarify more often why you are focusing on specific sites. Further, please address what biases the imperfect sampling scheme may cause in your analysis and how you address it. If space allows, also consider moving table S1 of sampling info into the main text.

In general, given the difference in timing of sampling, sampling effort, climate, and sequencing batches for each plot, more detail is needed in the reporting of the results to clarify how sampling ‘design’ influences methodology and interpretation.

Include a supplementary table of sequencing batch information and clarify in the text how you can be sure batch effects are not driving any of your results (e.g., clarify that PCA clusters on PCs 2-4 do not align with sequencing batch (?) to ensure they are biologically relevant clusters).

As written, there are several places where results give the impression of avoiding discussion of data that does not support the central arguments. I have left line-specific comments below about where this is most notable in the results. Beyond a lack of clarity about missing data from certain sites or years mentioned above, the discussion of CAC_S2 data (and LM data) is minimal. This feels misleading and seems like a missed opportunity to clarify what about these two streams causes different admixture and population structure outcomes.

The authors claim to address how the genetic structure of admixed populations is maintained over time, but their data do not directly address this question. They clearly outline a hypothesis in the discussion as a future direction - investigate microsite differences that may be responsible for differences in phenology and, thus, flowering time overlap and admixture. However, this paper does not address this directly, and any such claims should be removed. The authors should instead focus on more thoroughly exploring the results of this study and the differences in data across sites that are poorly addressed. For example, I don't believe the data from LM are even mentioned in the discussion.

Minor comments:

In the paragraph starting on line 125, you do a good job explaining how M. nasutus habitat and enviromental fluctiations impact flowering time and life history, but lack a comparison to M. guttatus. I think it is important to clarify here (with citations) how this impacts phenological overlap with M. guttatus as this is critical background for the reader to understand your conclusions.

Lines 139-141: This sentence is unclear. For example, it is not clear what you mean by “three additional years”, and I am left wondering whether sampling was different at the two sites. Why is one site named and the other not? Why did you do additional sampling at a ‘nearby site’? How do you know they were offspring - were they tracked in the wild, or grown in a greenhouse? If greenhouse grown, are you breeding them in the greenhouse, or collecting new seed from the field each year? While these details are in the materials/methods, given the results first format of this journal, you need to better clarify the experimental/sampling design up front for readers who may not read your methods first to be able to understand your results. Further, this description is hard to reconcile with the caption of Figure 1, where you define 3 sites. Please make sure you are consistent with how you describe the number of sites and sampling effort.

Methods:

520-521: It is not clear what is meant by the parenthetical here, especially the “1-3 individual fruits per maternal family” piece, even after looking at supplementary table S1. Please make the sampling/sequencing design more clear/explicit.

542 states that samples were sequenced across 3 runs but does not reference a table showing which samples were in the same run. This can be important for batch effects if one run contains all of one species or sampling location, etc. Please include information on what samples were grouped in the same sequencing batch and whether you see clustering in PCAs by batch (eg do samples of the same species ever cluster by batch), and how batch effects might influence your data interpretation.

544-549: please report standard genome coverage metrics (breadth and depth of coverage) rather than raw read numbers, which are not meaningful without reference to genome size, etc.

Starting on line 565, translate filters into plain English. Readers should go to the co-published code for the specific parameters (and that GitHub repository should be cited in the methods).

571-576: Please further clarify the differences between these two panels and why they were created. Are many of the sites in the second ancestry informative panel not covered in at least 60% of your test samples?

The inclusion of a test for the ancestry calling (lines 608-614) is a well-executed and important check! Nice work.

665-668: please state how many were excluded (not just how many were kept) to give a clear picture of how frequent this was. It would also be nice to know why the authors think this happened as if it is a substantial number, it might indicate that mix-ups happened in sampling/sequencing that could mean mismatched families and offspring with matching parent types were kept (ie looked fine in terms of coming from the right parents, but might be mislabeled in terms of family). The authors should clarify this point carefully to assure readers that the retained data are trustworthy.

681-681: Explain why this was only done for CAC_S1 (lack of 3 distinct cohorts, yes, but there is admixture that could be explored in CAC_S2, eg you could still quantify RI between the two species)

694-696: why was only one representative plot from one site chosen to represent the three groups? It seems like the phenological value should be calculated from data from all individuals in each cohort as was done for mating system. It currently gives the impression that those plots could have been cherry-picked for clear differences in flowering time. Either give a clear justification for only calculating phenological RI across those 3 plots (maybe because of the best consistent sampling as implied in the results?), or do the analysis across the full set of individuals per cohort (or at least all individuals per cohort in CA_S1 where flowering time is consistent within cohort across plots).

Line 712: This is a great aspect of your study; using offspring to look at assortative mating is cool and interesting!

724: If you see batch effects in a PCA (clustering of samples by library or sequencing run), you should include batch as a random variable. (true for all regressions in your analyses)

727: “for only families with maternal HI<0.5.” Remind readers what this means (what does HI<0.5 indicate)

Results:

170-171: Please add information about how this relates to physical distance as you did with the description of physical distance and genetic distance for CAC-1. Also, it seems possible that some of this structure could be related to batch effects given the relatively low variance explained by PC axes 2-4 and the lack of differnetiation between the 2 CAC streams when there is differentiation btw plots from a single stream. Again, please address this by clarifying which samples were in the same library preps/sequencing runs so it is clear that PCA clusters are not simply aligning with sequencing batches.

200: Why I this surprising? If you are going to see the nasutus/sookensis haplotype anywhere in guttatus, wouldn’t you expect it to be in a sympatric site?

212-215: You report the same HI peak ancestry for the CAC-S2 guttatus cohort and the CAC_S1 admixed cohort; this is also clear in figure 2C. How can the same hybrid ancestry be called the ‘admixed’ cohort in one location and guttatus in the other? I understand calling 3 different cohorts within CAC_S1, but ancesrty cohorts should be defined by consistent ancestry proportions, and therefore the CAC-S2 peak at HI .25-.3 should be called part of an admixed cohort.

215-216/219-220: Please clarify that this is across years for the 2 streams for which you had multiple years of data, and that you cannot confirm the patterns have been stable across years at the LM site. The current writing implies that your results show consistency across all three streams through time.

217-218: Here you contradict the previous statement and properly label the second CAS-S2 cohort as admixed. Please just adjust lines 212-215 to match this classification.

221-230: This is cool and interesting! However, please clarify why you are only looking at CAC_S1 (presumably because there is no spatial structure in S2). I think the study would be strengthened overall by a more thorough treatment of both streams and how they differ. Focusing on how and why you think these streams differ would be a more honest interpretation of your data than building a story about stable ancestry mosaics through time due to microhabitat variance when you do not have data on microhabitat. You do have lots of great genetic data, from multiple sites! Focus on this and fully interpreting it.

233: Again, interesting approach and results. However, here you go back to using data from both streams, even though the question being tested is based entirely on results from one stream. The reasons behind these choices of when to include different sites in the analysis need to be better clarified throughout.

242: Rather than ‘a small number’ explicitly state how many

306-307: This really is an awesome component of your study!

Discussion:

359-361: This logic feels at odds with your data/results; your plots vary in ancestry (mosaic of ancestry), but your results do not actually investigate the mechanisms underlying this differentiation. You do not look at microhabitat differences (moisture etc), and do not calculate RI across all plots, so you can’t make a statement about fluctuating RI and its relationship to admixture across the landscape. Further, the plots at the second stream, CAC_S2, presumably do not vary in ancestry across the same spatial scale, though this result was largely ignored. Your more nuanced explanation on 390-394 is great and clarifies that this (very reasonable) speculation about causation is grounded in the existing literature, not your results. Please revise this early statement and more clearly connect it with your actual data/results.

427-429: This statement would be bolstered by a more careful explanation of results from CAC_S2. If distances are the same but there is no population structure in S2 compared to S1, this also indicates that factors other than distance are responsible for the (admixture based) population structure in CAC_S1.

433-435: This feels misleading given that you did not explore the differences in microsite conditions and how they relate to differences in reproductive isolation, which is what you are describing as the factors likely responsible for a consistent mosaic of admixture.

Figures:

Figure 1B - At a minimum, please explain in the legend that colored lines are collection years, and grey is other years for context about precip variability. However, I recommend coloring and labeling all years, as in some cases, the environmental conditions of the preceding year are very important (reflecting selection on parents and produced seeds).

Figure 2 - Please clarify that LM and NAS were only removed visually, but were still part of the PC analysis that generated the PC3 and PC4 values.

In fact, if the point of showing PCs 3 and 4 is to examine patterns in just CAC, and visually exclude LM and NAS, you should do a separate PCA of CAC so that variance explained by the PCs is meaningful for understanding within CAC divergence.

You can include PC3 and PC4 for context in the supplement about how much pop variance there is in CAC relative to divergence with NAS, but since that is not a primary/important component of your results, I would recommend 2 separate PCAs instead. At the least, the PCA without NAS and LM should be run to ensure structure within CAC is consistent with the structure indicated by PC3 and PC4 in the larger analysis. It is also unclear in this why you specify in the legend (and not the figure itself) that LM-A was removed (what is LM-A and how is it subsetted from LM?)

Also, if you are going to include the allopolyploid, you need to explain why - what does inclusion add? What are the progenitor species, etc. (state in legend that it is allop btw nasutus and guttatus)

2B Is a nice visualization of the directional admixture!

2C - It looks like lower admixture in CAC-S2 could be largely due to demography and there being fewer overall nasutus. Please address this - is there less nasutus at S2?

It is confusing trying to follow which species/sites were sampled in which years. Please rewrite the legend and/or clarify with a visual like a small table showing site and the years with data for that site.

Figure 3: not everyone will be familiar with a NJ-Net Haplotype Network; please describe in the legend what each circle and its size represent, as well as the meaning of the labels and the hash marks on joining lines. This may sound silly/seem obvious, but it will help unfamiliar readers orient more easily to the figure and the key finding it shows.

Figure 4: It would be more intuitive for the labels to go from lowest number at the top rather than largest (S1-1 to S1-6 rather than visa versa). Why is S1-1 divided into A and B, but not the others?

The legend makes it sound like some plots are purely guttatus, some are admixed, and some are purely nasutus. Consider rephrasing for clarity, somethign like, “plots X, Y, and Z with primarily nasutus indivisuals have earlier flowering times”.

4 - The legend states that pops with admixed individuals have later flowering (intermediate) than M. nasutus, and that M. guttatus has late flowering. While this appears true in 2022, it does not appear completely accurate across all years (eg. 2019, S1-1 is later than S1-2, S1-5B was late in 2012, etc). Your other plots and analyses show this relationship more clearly, let those speak for themselves.

4D - Dotted lines are nearly impossible to see; please specify quantiles for the interquartile range (is it the standard 25th-75th percentile?).

4D/E - These are much more convincing plots for making the point alluded to in the description of 4B! Remove commentary from 4B in favor of these analyses.

4E - It looks like the relationship is different for CAC-1 and 2 in 2022, where the relationship seems weaker for CAC-2. It is misleading to lump CAC-1 and 2 for 2022 when you also plot 2019, but only CAC-1. Please either include separate trend lines for CAC 1 and 2. This plot and the claim of association between hybrid index and open flower date needs statistical support - please include a reference to the stats in the legend. Also, include a confidence interval around trend lines to visually represent the variance in the trend.

Figure 5 - The labeling within the plot is very helpful here! The first part of the second sentence in the legend is confusing. B is cool and interesting, but the blue and orange confidence interval shading are difficult to see. Please darken the shading in those colors or add a boundary to the shaded area.

**Have all data underlying the figures and results presented in the manuscript been provided?**

Reviewer #1: Yes

Reviewer #2: Yes

PLOS authors have the option to publish the peer review history of their article (what does this mean? ). If published, this will include your full peer review and any attached files.

**Do you want your identity to be public for this peer review?** For information about this choice, including consent withdrawal, please see our Privacy Policy .

Reviewer #1: No

Reviewer #2: **Yes: ** Julia Harenčár

---

## [Decision Letter · Decision Letter 1]

16 Feb 2025

Dear Dr Farnitano,

We are pleased to inform you that your manuscript entitled "Fluctuating reproductive isolation and stable ancestry structure in a fine-scaled mosaic of hybridizing *Mimulus*  monkeyflowers" has been editorially accepted for publication in PLOS Genetics. Congratulations!

Yours sincerely,

Jesse Lasky

Academic Editor

PLOS Genetics

Justin Fay

Section Editor

PLOS Genetics

Aimée Dudley

Editor-in-Chief

PLOS Genetics

Anne Goriely

Editor-in-Chief

PLOS Genetics

Comments from the reviewers (if applicable):

The authors have done a commendable job responding to previous reviewer comments.

Reviewer's Responses to Questions

**Comments to the Authors:**

Reviewer #1: The authors have completed a pretty nice and interesting research on reproductive isolation and hybridization by providing solid evidence. The revision is also satisfying.

Reviewer #2: The authors have done a great job addressing reviewer comments! I have no further comments and congratulate the authors on a job well done.

**Have all data underlying the figures and results presented in the manuscript been provided?**

Reviewer #1: Yes

Reviewer #2: None

PLOS authors have the option to publish the peer review history of their article (what does this mean? ). If published, this will include your full peer review and any attached files.

**Do you want your identity to be public for this peer review?** For information about this choice, including consent withdrawal, please see our Privacy Policy .

Reviewer #1: **Yes: ** Jun Chen

Reviewer #2: No

**Data Deposition**

http://datadryad.org/submit?journalID=pgenetics&manu=PGENETICS-D-24-01091R1

**Press Queries**

---

## [Editor Report · Acceptance letter]

PGENETICS-D-24-01091R1

Fluctuating reproductive isolation and stable ancestry structure in a fine-scaled mosaic of hybridizing *Mimulus*  monkeyflowers

Dear Dr Farnitano,

We are pleased to inform you that your manuscript entitled "Fluctuating reproductive isolation and stable ancestry structure in a fine-scaled mosaic of hybridizing *Mimulus*  monkeyflowers" has been formally accepted for publication in PLOS Genetics! Your manuscript is now with our production department and you will be notified of the publication date in due course.

With kind regards,

Anita Estes

PLOS Genetics

On behalf of:
